



# Implementation of a chemical background method for atmospheric OH measurements by laser-induced fluorescence: characterisation and observations from the UK and China

Robert Woodward-Massey[1,a], Eloise J. Slater[1], Jake Alen[1], Trevor Ingham[1,2], Daniel R. Cryer[1], Leanne M. Stimpson[1], Chunxiang Ye[1,a], Paul W. Seakins[1], Lisa K. Whalley[1,2] and Dwayne E. Heard[1]

[1]School of Chemistry, University of Leeds, Leeds, LS2 9JT, UK
[2]National Centre for Atmospheric Science, University of Leeds, Leeds, LS2 9JT, UK
[a]now at: College of Environmental Sciences and Engineering, Peking University, Beijing, 100871, China

*Correspondence to*: L. K. Whalley (l.k.whalley@leeds.ac.uk) and D. E. Heard (d.e.heard@leeds.ac.uk)

**Abstract.** Hydroxyl (OH) and hydroperoxy (HO$_2$) radicals are central to the understanding of atmospheric chemistry. Owing to their short lifetimes, these species are frequently used to test the accuracy of model predictions and their underlying chemical mechanisms. In forested environments, laser-induced fluorescence–fluorescence assay by gas expansion (LIF–FAGE) measurements of OH have often shown substantial disagreement with model predictions, suggesting the presence of unknown OH sources in such environments. However, it is also possible that the measurements have been affected by instrumental artefacts, due to the presence of interfering species that cannot be discriminated using the traditional method of obtaining background signals via modulation of the laser excitation wavelength ("OHwave"). The interference hypothesis can be tested by using an alternative method to determine the OH background signal, via the addition of a chemical scavenging prior to sampling of ambient air ("OHchem"). In this work, the Leeds FAGE instrument was modified to include such a system to facilitate measurements of OHchem, in which propane was used to selectively remove OH from ambient air using an inlet pre-injector (IPI). The IPI system was characterised in detail, and it was found that the system did not reduce the instrument sensitivity towards OH (<5% difference to conventional sampling), and was able to efficiently scavenge external OH (>99%) without the removal of OH formed inside the fluorescence cell (<5%). Tests of the photolytic interference from ozone in the presence of water vapour revealed a small but potentially significant interference, equivalent to an OH concentration of ~4 × 10$^5$ molecule cm$^{-3}$ under typical atmospheric conditions of [O$_3$] = 50 ppbv and [H$_2$O] = 1%. Laboratory experiments to investigate potential interferences from products of isoprene ozonolysis did result in interference signals, but these were negligible when extrapolated down to ambient ozone and isoprene levels. The interference from NO$_3$ radicals was also tested but was found to be insignificant in our system. The Leeds IPI module was deployed during three separate field intensives that took place in summer at a coastal site in the UK, and both in summer and winter in the megacity Beijing, China, allowing for investigations of ambient OH interferences under a wide range of chemical and meteorological conditions. Comparisons of ambient OHchem measurements to the traditional



OHwave method showed excellent agreement, with OHwave vs OHchem slopes of 1.05–1.16 and identical behaviour on a diel basis, consistent with laboratory interference tests.

## 1 Introduction

The removal of pollutants and greenhouse gases in the troposphere is dominated by reactions with the
hydroxyl radical (OH), which is closely coupled to the hydroperoxy radical ($HO_2$). Comparisons of the levels of OH and $HO_2$ observed during field campaigns to the results of detailed chemical box models serve as a vital tool to assess our understanding of the underlying chemical mechanisms involved in tropospheric oxidation. Laser-induced fluorescence–fluorescence assay by gas expansion (LIF–FAGE) measurements of OH in forested environments have often been considerably higher than those predicted by models (Carslaw
et al., 2001; Lelieveld et al., 2008; Ren et al., 2008; Hofzumahaus et al., 2009; Stone et al., 2011; Whalley et al., 2011; Wolfe et al., 2011). The difficulty in simulating radical concentrations in such environments has prompted a multitude of theoretical (Peeters et al., 2009; da Silva et al., 2010; Nguyen et al., 2010; Peeters and Muller, 2010; Peeters et al., 2014), laboratory (Dillon and Crowley, 2008; Hansen et al., 2017), and chamber (Paulot et al., 2009; Crounse et al., 2011; Crounse et al., 2012; Wolfe et al., 2012; Fuchs et al., 2013;
Fuchs et al., 2014; Praske et al., 2015; Teng et al., 2017; Fuchs et al., 2018) studies to help explain the sources of the measurement–model discrepancy, through detailed investigations of the mechanism of isoprene oxidation under low $NO_x$ conditions, as well as other biogenic volatile organic compounds (BVOCs) (Novelli et al., 2018). However, another hypothesis is that the LIF measurements have, at least in part, suffered from an instrumental bias in these environments due to interfering species.

Early LIF measurements of OH suffered from significant interferences due to laser-generated OH from ozone photolysis in the presence of water vapour (Hard et al., 1984). While this effect has been reduced in going from 282 to 308 nm laser excitation of OH, it may still be significant, especially at night or with the use of multi-pass laser setups (e.g., up to ~$4 \times 10^6$ molecule $cm^{-3}$ in Griffith et al. (2016)). Laboratory experiments conducted by Ren and co-workers using the Pennsylvania State University (PSU) LIF instrument
showed negligible interferences in OH detection for a range of candidate species: $H_2O_2$, HONO, HCHO, $HNO_3$, acetone, and various $RO_2$ radicals (Ren et al., 2004). Observations of OH during the PROPHET (Program for Research on Oxidants: PHotochemistry, Emissions and Transport) field campaign in summer 1998, located in a mixed deciduous forest in Michigan, USA, revealed unusually high nighttime OH concentrations (~$1 \times 10^6$ molecule $cm^{-3}$) but measurement interferences were ruled out (Faloona et al., 2001).

However, the results of more recent studies conducted in forested environments have meant that interferences in the measurement of OH by LIF–FAGE have been revisited. The usual background method of this technique, where the laser wavelength is scanned off-resonance from an OH transition ("OHwave"), does not discriminate between ambient (i.e., "real") OH and either OH formed inside the FAGE cell (e.g., laser- or surface-generated OH, or via unimolecular decomposition in the gas phase to form OH), or
fluorescence from other species at $\lambda \sim 308$ nm (e.g., naphthalene, $SO_2$), although it is possible to correct for such effects providing the interference has been previously characterised (Martinez et al., 2004; Ren et al.,



2004; Griffith et al., 2013; Fuchs et al., 2016). There is an alternative, chemical method ("OHchem") for obtaining the OH background signal in LIF instruments that allows for interference signals to be determined without their prior characterisation, to rule out possible artefacts from unknown species. The OHchem

method involves the addition of a high concentration of an OH scavenger, such as perfluoropropene ($C_3F_6$) or propane, just before the FAGE inlet. Ambient OH is quickly titrated away by fast reaction with the scavenger, but any interference should remain in the fluorescence signal, although this must be corrected for reaction of any internally generated OH with the scavenger inside the FAGE cell.

Several LIF–FAGE groups have now made efforts to validate ambient OH measurements through

incorporation of the alternative OHchem technique, which was first applied for continuous OH measurements by (Mao et al., 2012). Since then, field studies of OH measurement interferences have been conducted in forested (Griffith et al., 2013; Novelli et al., 2014a; Feiner et al., 2016; Novelli et al., 2017), rural (Fuchs et al., 2017; Tan et al., 2017), suburban (Tan et al., 2018; Tan et al., 2019), urban (Ren et al., 2013; Brune et al., 2016; Griffith et al., 2016), and coastal (Novelli et al., 2014a; Mallik et al., 2018) locations. Substantial

improvement in measurement–model agreement has been possible when OH backgrounds were determined chemically, especially in forested environments, suggesting that understanding of tropospheric oxidation processes in such regions may be better than previously thought (Mao et al., 2012; Hens et al., 2014; Feiner et al., 2016). This is further supported by the positive identification of two new OH interference candidates in laboratory experiments, namely intermediates in alkene ozonolysis reactions, which may (Novelli et al.,

2014b; Novelli et al., 2017; Rickly and Stevens, 2018) or may not (Fuchs et al., 2016) be related to stabilised Criegee intermediates (SCIs), and the $NO_3$ radical (Fuchs et al., 2016) although for all cases the observed interferences cannot explain the magnitudes of the OH background signals under ambient conditions. The trioxide species, ROOOH, has also been postulated to explain elevated OH backgrounds in LIF–FAGE measurements made in forested regions (Fittschen et al., 2019).

However, it is not known whether other LIF instruments suffer the same levels of interference, which are likely highly dependent on cell design and operating parameters, especially the residence time of air between sampling and detection (Novelli et al., 2014a; Fuchs et al., 2016; Rickly and Stevens, 2018). Considering the bespoke nature of LIF–FAGE instruments, those of different groups share the same main features but differ in many aspects, such as inlet size and shape, or whether the laser crosses the detection axis once (i.e., single-

pass) or multiple times (multi-pass). As a result, the magnitude of any interference is likely to vary significantly between different instruments. Because of this, a general recommendation of the 2015 International $HO_x$ Workshop (Hofzumahaus and Heard, 2016) was that different groups should incorporate their own chemical scavenger system for use in ambient OH measurements, and to test interferences in the laboratory.

Following this recommendation, the Leeds ground-based FAGE instrument was modified to incorporate a chemical scavenger system, through the addition of an inlet pre-injector (IPI). In this work, we describe the design of the IPI system and its thorough characterisation in terms of sensitivity and the degree of external and internal OH removal. Following this, we present the results of interference testing experiments performed



using the IPI system, in which we investigated interferences from $O_3 + H_2O$, isoprene ozonolysis, and $NO_3$
radicals. Finally, we demonstrate the use of the optimised IPI system for measurements of ambient OH made
during three separate field campaigns in the UK and China, which encompassed a wide range of chemical
and meteorological conditions.

## 2 Methodology

### 2.1 Overview of the Leeds ground-based FAGE instrument

The University of Leeds ground-based FAGE instrument, described in detail elsewhere (Creasey et al.,
1997a; Whalley et al., 2010; Whalley et al., 2013), has participated in 24 intensive field campaigns since its
initial deployment in 1996. Measurements of OH, $HO_2$, and, more recently, $RO_2$ radicals (Whalley et al.,
2013), have been made in a variety of locations, ranging from pristine marine boundary layer (Creasey et al.,
2003; Whalley et al., 2010), tropical rainforest (Whalley et al., 2011), and polar (Bloss et al., 2007)
environments, to coastal (Smith et al., 2006) and semi-polluted regions (Creasey et al., 2001), as well as
urban areas (Heard et al., 2004; Emmerson et al., 2007) including a highly polluted megacity (Lee et al.,
2016; Whalley et al., 2016).

Ambient OH is measured using laser-induced fluorescence. Briefly, ambient air is drawn through a 1.0
mm diameter pinhole in a conical turret inlet (4 cm length, 3.4 cm ID; Figure 1) at ~7 slm into a stainless
steel fluorescence cell, held at ~1.5 Torr using a Roots blower (Leybold RUVAC WAU 1001) backed by a
rotary pump (Leybold SOGEVAC SV200). An all solid-state laser system is used to excite OH via the $A^2\Sigma^+(v'
= 0) \leftarrow X^2\Pi_{3/2}(v'' = 0)$ electronic transition at $\lambda = 308$ nm ($HO_2$ is measured by conversion to OH using NO,
details below). The resultant fluorescence at 308 nm is detected by a micro-channel plate photomultiplier
(MCP, Photek PMT325/Q/BI/G with 10 mm diameter photocathode) equipped with a 50 ns gating unit
(Photek GM10-50) and a 2 GHz 20 dB gain amplifier (Photek PA200-10), and the signal analysed by gated
photon counting (Whalley et al., 2010). The background signal is normally obtained by scanning the laser
wavelength off-resonance from the OH transition line, yielding the measurement commonly referred to as
OHwave:

$$[OHwave] = C_{OH} \times (S^{OH}_{online} - S^{OH}_{offline})$$
$$= C_{OH} \times S^{OH} \tag{1}$$

where $C_{OH}$ is the instrument calibration factor for OH, and $S^{OH}_{online}$ and $S^{OH}_{offline}$ are the OH LIF signals at on-
and off-resonance wavelengths, respectively. Similarly, the alternative measurement known as OHchem is
defined as:

$$[OHchem] = C_{OH} \times (S^{OH}_{online} - S^{OH}_{online, scavenger})$$
$$= C_{OH} \times S^{OH}_{scavenger} \tag{2}$$

where $S^{OH}_{online,\ scavenger}$ is the OH signal measured at an on-resonance wavelength but in the presence of a

scavenger.

HO$_2$ is detected via its conversion to OH following the addition of NO (BOC, 99.95% and Messer, 99.95%). Although not reported here, RO$_2$ radicals are measured using the RO$_x$LIF method (Fuchs et al., 2008; Whalley et al., 2013), in which their reactions with NO and CO (BOC, 5% in N$_2$ and Messer, 10% in N$_2$) result in conversion initially to OH and subsequently to HO$_2$ that is then detected as described above.

The NO and CO are delivered using mass flow controllers (MFC, MKS Instruments 1179A series), which, unless otherwise stated, were also used to control all other gas flows described in this work. The Leeds FAGE instrument features two fluorescence cells, where laser light (~10–20 mW at 308 nm, supplied at a pulse repetition frequency of 5 kHz via an optical fibre) enters each cell in series. For normal operation in the field, the first cell (HO$_x$) measures OH and HO$_2$ (low NO flow, 5 sccm; RO$_2$ interference minimised) sequentially,

while the second cell (RO$_x$) measures HO$_2$* (high NO flow, 50 sccm; RO$_2$ interference maximised) and then total RO$_2$ (Fuchs et al., 2008; Whalley et al., 2013).

Calibration of the FAGE instrument is achieved by supplying known radical concentrations via a turbulent flow tube (known in Leeds as the "wand") held at ~45° to the instrument inlet, where OH and HO$_2$ are formed in a 1:1 ratio (Fuchs et al., 2011) by the photolysis of water vapour at 184.9 nm using a Hg(Ar) pen-ray lamp

(LOT LSP035) in an excess flow (40 slm) of humidified ultra-high purity air (BOC, BTCA 178 and Messer, 20.5% O$_2$ in N$_2$). Chemical actinometry is performed via the photolysis of N$_2$O (BOC, medical grade 98%) to measure the product of lamp flux and photolysis exposure time to enable calculation of radical concentrations (Edwards et al., 2003; Faloona et al., 2004). The calibration of OH using the water vapour photolysis method has been validated by comparison with other methods, for example the kinetic decay of

hydrocarbons (Winiberg et al., 2015).

## 2.2 Inlet pre-injector (IPI) design

The Leeds inlet pre-injector (Figure 1) is similar in concept to the design of Mao et al. (2012) and consists of a 4.0 cm length, 1.9 cm ID perfluoroalkoxy (PFA) cylinder embedded inside an aluminium housing, which seals to the FAGE cell via an O-ring base. The scavenger is injected into the centre of the PFA flow tube via

four 0.25 mm ID needles, 4.0 cm above the FAGE inlet. The low bore capillary tubing increases the pressure inside the needles, which facilitates mixing of the scavenger into the ambient air stream. In this work propane (BOC, research grade 99.95% and Messer, 99.995%) was used as the main OH chemical scavenger, with similar results (see section 3.1.2) obtained for C$_3$F$_6$ (Sigma-Aldrich, 99%). Based on previous investigations of OH interferences (Stevens et al., 1994; Dubey et al., 1996; Faloona et al., 2001; Ren et al., 2004; Mao et

al., 2012; Griffith et al., 2013), C$_3$F$_6$ was used initially as it reacts quickly and selectively with OH ($k_{298} = 2.2 \times 10^{-12}$ cm$^3$ molecule$^{-1}$ s$^{-1}$ (Sander et al., 2011)), and does not contain any hydrogen atoms which could serve as a source of laser-generated OH via abstraction by O($^1$D) atoms (Stevens et al., 1994; Dubey et al., 1996). However, C$_3$F$_6$ must be diluted in an inert gas before it can be flowed through MFCs, and its availability in the UK became limited in 2015. Following Novelli et al. (2014a), we therefore used propane for most



laboratory experiments and all ambient measurements, despite the fact that it reacts more slowly with OH ($k_{298} = 1.1 \times 10^{-12}$ cm$^3$ molecule$^{-1}$ s$^{-1}$ (Sander et al., 2011)).

As shown in Figure 2, the scavenger (0–50 sccm) is diluted in a flow of $N_2$ (500 sccm, BOC, 99.998%) prior to injection, which is controlled using a solenoid valve (Metron Semiconductors). Any dead volume after the valve is purged continuously by the $N_2$ dilution flow, using a narrow-bore injector inserted through

the tee after the valve, with the injector tip placed as close to the valve orifice as possible. This enables fast flushing of the system to optimise the response time before and after scavenger injection. Incorporation of the purge system resulted in pre- and post-injection stabilisation times on the order of seconds (data not shown), minimising data loss. The valve state and scavenger flow over the course of the data acquisition cycle are controlled using a custom program nested within the FAGE software.

To reduce radical wall losses, excess ambient air is drawn through the IPI to generate a sheath flow, via four ports spaced evenly around the flow tube housing as shown in Figure 1. This minimises the FAGE sampling of air from near the walls of the cylinder, housing, and turret. The total flow rate through the IPI is 32 slm, of which 7 slm is sampled by the FAGE cell and the remainder of the flow is maintained by a vacuum pump (Agilent Technologies IDP-3 Dry Scroll Pump) and measured volumetrically using a rotameter

(Brooks 2520, 4–50 L min$^{-1}$).

During interference testing experiments using the IPI system (section 3.2), ozone and water vapour concentrations were measured using a commercial UV absorption instrument (Thermo Environmental Instruments Inc. 49C) and a chilled mirror dew point hygrometer (General Eastern 1311DR sensor and 4×4 Optica), respectively.

**2.3 Field measurement sites**

Ambient measurements of OHwave and OHchem were made using the Leeds IPI–FAGE instrument during three separate intensive field campaigns, in different locations and seasons. This allowed for the investigation of potential OH interferences under markedly contrasting conditions. For all three field campaigns, measurements of OH, $HO_2$, and partially speciated $RO_2$ were made using the Leeds FAGE instrument (4 m

above ground level), operated in the sequential detection modes described in section 2.1 (Whalley et al., 2013). Total OH reactivity, $k'_{OH}$, was also measured, using the laser flash photolysis–LIF instrument described in detail by (Stone et al., 2016). A range of supporting chemical, aerosol, and meteorological parameters were measured, with instruments situated either in buildings or shipping containers at each of the two sites. Gas phase chemical observations included water vapour, $NO_x$, $NO_y$, $O_3$, CO, $SO_2$, HONO, HCHO

(Cryer, 2016), $ClNO_2$, VOCs, and OVOCs. Photolysis rates ($J$) for a variety of species, including $O_3$ ($\rightarrow$ O($^1$D)), $NO_2$, HCHO, HONO, and $ClNO_2$, were measured using a $2\pi$ spectral radiometer ($2\pi$ actinic receiver optic (Meteorologie Consult GmbH) coupled to an Ocean Optics QE Pro spectrometer), and $J$(O$^1$D) was also measured using a $2\pi$ filter radiometer (Meteorologie Consult GmbH) (Bohn et al., 2008). The meteorological and chemical conditions, including some example VOCs, encountered during each campaign are summarised

in Table 1 and discussed in further detail below.



The first deployment of the Leeds IPI was during the ICOZA (Integrated Chemistry of OZone in the Atmosphere) project, which took place in July 2015, at the Weybourne Atmospheric Observatory (WAO), Weybourne, located on the North Norfolk Coast, UK (52°57'02''N, 1°07'19''E, 15 m asl). The WAO is a Global Atmospheric Watch (GAW) Regional station, and the site is impacted by a range of contrasting air

masses, from clean Arctic air to processed emissions from the UK (e.g., London, which is located ~180 km SSW of the observatory) and Northern Europe. The aim of this field campaign was to improve understanding of ozone chemistry through integrated measurements of $P(O_3)$, the chemical or *in situ* ozone production rate (OPR) (Cazorla and Brune, 2010; Cazorla et al., 2012), with comparisons to a range of other observational and model approaches.

In general, the ICOZA campaign was characterised (Table 1) by moderate temperatures (16 °C median), high humidity (RH ~ 80%), and strong wind speeds (~6 m s$^{-1}$), as might be expected at a temperate, coastal location in the summertime. The predominant wind sector, based on wind direction measurements at the site, was westerly (~30%), followed by southwesterly (~20%) and southerly (~15%). Back-trajectory analysis showed that during IPI sampling periods, the site was predominantly under the influence of Atlantic air

(Cryer, 2016). These air masses had spent a considerable amount of time (~1 day) over the UK, often encountering emissions from urban areas, which underwent photochemical aging during their transport to the WAO site. Overall, the levels of pollution observed at the site were moderate, and the lowest of the three field campaigns discussed in this work (Table 1), and levels of isoprene were low. However, ozone mixing ratios were relatively high, with a diel-average maximum of ~40 ppbv, driven in part by strong UV and near-

UV radiation.

The Leeds IPI was deployed during another two campaigns at the Institute of Atmospheric Physics (IAP, 39°58'36.06''N, 116°22'53.69''E), an urban site in Beijing, China, during winter (November–December) 2016 and summer (May–June) 2017, as part of the AIRPRO (an integrated study of AIR pollution PROcesses in Beijing) project. AIRPRO is part of the wider APHH (Air Pollution and Human Health in a Chinese

megacity) project (Shi et al., 2018), a joint UK–China programme. The aims of AIRPRO included the assessment of how pollutants are transformed and removed through transport, chemical, and photolytic processes, with a particular emphasis on the identification of the dominant oxidative degradation pathways (i.e., the relative importance of reactions with OH, NO$_3$, and O$_3$). The AIRPRO project allowed for the assessment of OH measurement interferences under the highly polluted conditions of the megacity Beijing,

situated in the heavily industrialised North China Plain. In winter, the site is impacted by urban and regional anthropogenic emissions, in particular those from the combustion of fossil fuels for residential heating. During summer, the site is subject to additional biogenic influences, and strong photochemical activity results in high rates of ozone production.

For both AIRPRO field intensives, the predominant wind sectors were westerly and

southerly/southeasterly, which generally result in higher pollutant concentrations (Chen et al., 2015). Indeed, the two campaigns were subject to high pollutant concentrations, as illustrated by the elevated levels of NO$_2$, CO, propane, benzene, and $k'_{OH}$, many of which were over an order of magnitude higher than ICOZA (Table



1). In addition, the biogenic influence during summer is clear from the relatively high isoprene concentrations observed, ~0.4 ppbv on average but reaching up to 7.9 ppbv, a level considerably higher than those observed

in some forested environments. Despite similar $J(O^1D)$ values between ICOZA and AIRPRO summer, the higher VOC loadings during the latter resulted in increased production of ozone (90 ppbv diurnally-averaged maximum). In contrast, AIRPRO winter was characterised by small ozone mixing ratios (15 ppbv diurnal maximum), as a consequence of high NO levels (median 22 ppbv) and weak UV radiation. In summer, NO levels were high in the morning (~14 ppbv at 06:00 China Standard Time (CST)) but surprisingly low in the

afternoon, with diel-average median levels of ~0.5 ppbv (15:00–18:00 CST).

For all ambient observation periods, the IPI data acquisition cycle consisted of 5 minutes of online wavelength and 30 seconds of offline wavelength (spectral background) integration, where the online period was split into 2 minutes of OH measurements and 2 minutes of propane addition to the IPI flow tube (chemical background), followed by 1 minute of $HO_2$ measurements (by the addition of NO to the FAGE cell). In terms

of instrumental operation, the only difference between ICOZA and the AIRPRO campaigns was the use of different propane flows in the IPI. The propane concentration in the IPI flow tube was ~110 ppmv ($k'_{OH}$ ~ 3000 $s^{-1}$, $\tau_{OH}$ ~ 0.3 ms) during ICOZA and AIRPRO winter but, after internal removal experiments revealed that the propane level could be increased further (see section 3.1.3), a ten-fold higher concentration (~1100 ppmv), resulting in a concomitant reduction in the OH lifetime, was used for the AIRPRO summer campaign.

On one day with high ozone (up to ~80 ppbv) and moderate isoprene (~0.5–1 ppbv) levels, the propane mixing ratio was reduced to ~110 ppmv, but this had no observable effect on the background signals obtained for the summer data. All ambient OHwave data presented here have been corrected for the known interference from $O_3$ in the presence of $H_2O$ vapour (see section 3.2.1).

## 3 Results

### 3.1 IPI characterisation

#### 3.1.1 Sensitivity

The presence of additional surfaces in the IPI system may result in radical wall losses and therefore reduce the overall FAGE instrument sensitivity. To test for potential OH losses in the IPI flow tube, OH radicals were generated using a 184.9 nm Hg lamp placed at ~19 cm from the instrument inlet, so that ambient air

with elevated radical concentrations (~2–7 x $10^7$ molecule $cm^{-3}$) was sampled, alternating between IPI and non-IPI sampling (Figure 3), where for the latter the entire IPI assembly was removed. The dominant source of OH was the photolysis of ambient water vapour at 184.9 nm. In these experiments, the Hg lamp was placed sufficiently far away from each inlet within a large tent enclosure on the container roof, such that it could be assumed that OH concentrations were uniform in the region the inlet sampled from. Otherwise, the difference

in inlet height between IPI and non-IPI sampling may have resulted in different OH concentrations being sampled, e.g., due to differences in $O_3$ absorption at 184.9 nm ($O_3$ has a high cross-section at this wavelength), which would affect the light flux at the point of sampling and hence the concentration of OH generated. Since



ambient variability (e.g., in $NO_x$ levels) also affects the atmospheric radical concentrations, the IPI/non-IPI cycle was repeated several tens of times on three different days within the tent enclosure to ensure sufficient

averaging of the results. Any differences in wind speed or direction during the different days are not important because of the tent enclosure. Based on the averages for each set of repeat measurements in Figure 3, these experiments yield a mean $\pm 2\sigma$ IPIoff/IPIon ratio of 1.043 ± 0.023, i.e., a <5% sensitivity reduction due to the presence of the IPI. While $HO_2$ loss was not tested, the relative sensitivity is assumed to be closer to unity since it is less reactive than OH. In either case, the correction is smaller than the total instrumental uncertainty

(~26% at $2\sigma$), and as such no corrections were applied to OH or $HO_2$ calibration factors for the final workup of ambient data collected during IPI sampling periods.

The lack of OH loss in the IPI system is further supported by another test conducted in the field during the summer 2017 AIRPRO campaign, where on one day of the campaign, sequential measurements of OHwave were taken with and without the IPI assembly present. While this was not a formal intercomparison,

the summer 2017 campaign provided ideal conditions to assess IPI losses, considering the very high radical concentrations observed (OH frequently >$1 \times 10^7$ molecule $cm^{-3}$) in Beijing and thus a good signal-to-noise ratio. The results of this experiment are shown for OH in Figure 4. It can be seen that if no correction for a reduction in sensitivity reduction is applied, adjacent IPIoff and IPIon periods of data are qualitatively in agreement, with no discontinuities in the temporal profile, implying that IPI sensitivity loss is close to zero

under field operating conditions. Similar results were obtained for $HO_2$ (data not shown).

### 3.1.2 External OH removal

The external OH removal efficiency in the IPI system is controlled by the injection height, the choice of scavenger (i.e., the rate coefficient of the reaction of scavenger with OH), the scavenger and $N_2$ dilution gas flows delivered to the injectors, as well as the sheath flow. A key requirement here is efficient mixing of the

scavenger into the ambient air stream, which is difficult considering the fast flow rate and hence short residence time of air in the IPI flow tube. Additionally, it is important to consider that some reaction of the scavenger may occur inside the fluorescence chamber (internal OH removal, section 3.1.3). This would give rise to a positive bias in ambient OH concentration measurements made using the OHchem method, as internal OH removal could result in loss of interfering OH and therefore an apparent reduction in the true

background signal.

External OH removal experiments were performed by supplying known concentrations of OH and $HO_2$ to the instrument using the calibration wand described in section 2.1. However, in contrast to normal calibration procedures, where the wand is held at 45° to the pinhole (to overfill the pinhole and minimise sampling of pockets of air which may have been in contact with the metal pinhole surface), IPI

characterisation experiments were performed with the wand positioned parallel to the direction of flow within the IPI (i.e., 90° relative to the plane of the pinhole), with a distance of ~3 cm between the wand exit and the PFA flow tube. The high flow through the calibration wand (40 slm) ensured that an excess of calibration gas was delivered to the IPI system (sample flow ~ 32 slm).



The external OH removal efficiency ($RE^{OH}_{external}$) may be calculated from the proportion of OH remaining

($R^{OH}_{external}$) after injection of the scavenger, obtained from the ratio of the OH signals in the presence ($S^{OH}_{scavenger}$) and absence ($S^{OH}$) of the scavenger:

$$R^{OH}_{external} (\%) = 100 \times S^{OH}_{scavenger} / S^{OH} \tag{3}$$
$$RE^{OH}_{external} (\%) = 100 - R^{OH}_{external} \tag{4}$$


Initial tests included variation of the $N_2$ dilution flow, however the OH removal efficiency was generally low (data not shown), likely due to poor mixing of the scavenger into the sampled air when the flow rate from the injector is small. As a result, the $N_2$ dilution was set to the maximum flow of the MFC used (0.5 slm) for all subsequent experiments. Any further dilution of the ambient air stream would result in a loss of sensitivity

towards the detection of radicals, however, at 0.5 slm the dilution flow is virtually negligible compared to the total flow rate in the IPI system (32 slm). In other preliminary experiments, the scavenger was injected closer to the FAGE inlet (1.0 and 2.5 cm), but this also resulted in poor external OH removal owing to the shorter residence time between scavenger injection and FAGE sampling.

The scavenging efficiency was determined for both propane and $C_3F_6$, with good agreement between the

two scavengers. Figure 5 shows the remaining OH signal as a function of the OH reactivity ($k'_{OH} = k_{OH+scavenger}$ [scavenger]) calculated in the flow tube, which normalises the scavenger concentrations according to their different reaction rates with OH. The observed removal efficiency is in broad agreement with the theoretical scavenging efficiency, based on the residence time in the flow tube (~20 ms, assuming plug flow) and assuming perfect mixing, suggesting that in the Leeds IPI system the scavenger is well mixed into the gas

sampled by the FAGE cell. An optimum removal of virtually 100% (OH remaining $\pm 2\sigma = 0.030 \pm 0.091\%$) was observed at $k'_{OH} \sim 3000$ s$^{-1}$, equivalent to ~110 ppmv ($2.7 \times 10^{15}$ molecule cm$^{-3}$) propane. This scavenger concentration was used for measurements of OHchem during the summer 2015 ICOZA project and winter 2016 AIRPRO project. For the summer 2017 AIRPRO project, a ten-fold higher scavenger concentration was used (~1100 ppmv propane), after internal removal experiments revealed no loss of internal OH at this

higher concentration, as discussed in detail in the next section.

### 3.1.3 Internal OH removal

Internal removal of OH was quantified by Mao et al. (2012) after forming OH inside the PSU ground-FAGE cell using a Hg lamp, and comparing the OH signal with and without the presence of the scavenger ($C_3F_6$), added externally in the IPI system. It was found that most of the internal removal occurred in the instrument

inlet, rather than in the OH detection axis, with a total loss of ~20%. Internal removal was not tested in the laboratory by Novelli et al. (2014a) for the Max Planck Institute for Chemistry (MPIC) FAGE instrument, but instead they limited the scavenger (propene and propane) concentration such that the external OH removal efficiency was < 95%, to minimise possible reaction of the scavenger with OH inside the fluorescence cell. However, during ambient, nighttime tests (constant atmospheric OH concentration assumed), no change in





the OH background signal was observed after increasing the scavenger concentration by a factor of seven, providing evidence for a lack of internal removal (Novelli et al., 2014a).

In the present study, a novel approach was devised to quantify internal removal of OH in the Leeds IPI–FAGE instrument. First, under otherwise identical experimental conditions to those for external OH removal tests, sufficient CO (75 sccm, 95 ppmv) was added to the calibration wand to verify that the OH formed

(alongside $HO_2$) from the photolysis of water vapour was almost quantitatively converted to $HO_2$ (98.0 ± 0.4%, data not shown). Secondly, in addition to the calibration wand CO flow, a high flow of NO (50 sccm) was injected inside the FAGE cell, with the injector tip positioned centrally just below the turret pinhole, to reconvert the $HO_2$ back to OH for LIF detection; these experimental conditions ensured a fairly high $HO_2$-to-OH conversion efficiency of approximately 30%. In this manner, OH was only generated inside the FAGE

cell, and not in the IPI flow tube, such that any change in the fluorescence signal could be attributed to internal reaction of OH with propane, rather than reaction in the flow tube. The procedure for determination of internal OH removal bears some resemblance to that used for ambient detection of $RO_2$ using the $RO_x$LIF technique (Fuchs et al., 2008; Whalley et al., 2013), i.e., the external conversion of all radical species to $HO_2$ before internal conversion to OH within the fluorescence cell.

The internal OH removal efficiency ($RE^{OH}_{internal}$) was quantified in an analogous manor to the external scavenging efficiency, using the total fluorescence signal in the presence ($S^{HOx}_{scavenger}$) and absence ($S^{HOx}$) of the scavenger:

$$R^{OH}_{internal} (\%) = 100 \times S^{HOx}_{scavenger} / S^{HOx} \qquad (5)$$

$$RE^{OH}_{internal} (\%) = 100 - R^{OH}_{internal} \qquad (6)$$

Figure 6 shows a time series of the LIF signal during two example internal removal experiments. Here, the LIF signal represents the sum of signals from OH and $HO_2$, since they are produced in a 1:1 ratio (Fuchs et al., 2011) in the calibration wand. For both of the propane mixing ratios used, which were shown to result in

near complete external OH removal in section 3.1.2, there was no obvious decrease in the LIF signal, indicating no significant internal removal of OH. The average $\pm 2\sigma$ internal OH removal observed for repeat experiments was $-0.5 \pm 1.3\%$ (Table 2) at a propane mixing ratio of ~110 ppmv (ICOZA and AIRPRO winter conditions). For repeat experiments at the higher propane mixing ratio used during the AIRPRO summer field campaign (~1100 ppmv), the internal removal was still very small, and almost insignificant ($2.8 \pm 2.3\%$,

Table 2). The observed internal OH removal may be compared to that which might be expected theoretically. In the ambient pressure flow tube, a propane mixing ratio of 1100 ppmv equates to $k'_{OH} = 30{,}000$ s$^{-1}$, but this is a factor of 760/1.5 lower in the detection cell (i.e., the ratio of ambient to cell pressure), 59 s$^{-1}$ (assuming constant gas density and no change in the OH + propane rate coefficient). Under normal operation, NO injection occurs 10.5 cm below the pinhole, and 7.5 cm away from the laser axis (i.e., total of 18 cm between

the pinhole and detection volume), with a residence time of 0.9 ms between NO injection and OH detection (Creasey et al., 1997b; Whalley et al., 2013). The gas likely slows down between pinhole sampling and NO

injection, but assuming a constant gas velocity, the residence time between the pinhole and the laser axis is estimated at ~2 ms. Based on this, an internal OH removal of efficiency ~12% is calculated, which is higher than observed, likely because the assumption of constant gas velocity is invalid (i.e., the real residence time

is closer to ~1 ms) and mixing between $HO_2$ and NO from the injector is not instantaneous. However, it should be noted that this calculation also neglects the fact that the density is higher in the jet, or the perturbation to normal flow caused by moving the NO injector close to the pinhole.

### 3.2 Interference testing experiments

### 3.2.1 $O_3$ + $H_2O$ vapour

In LIF–FAGE instruments, there is a known interference due to laser-generated OH from ozone photolysis in the presence of water vapour (Fuchs et al., 2016; Griffith et al., 2016; Tan et al., 2017). This interference was quantified by (Whalley et al., 2018) and characterised in further detail in the present work. In these experiments, ozone was generated from the 184.9 nm photolysis of oxygen in a 12–20 slm flow of zero air using a Hg(Ar) pen-ray lamp (LOT LSP035). Another 12–20 slm of zero air was humidified using a water

(HPLC grade) bubbler. The two zero air flows were combined and delivered to the calibration wand, from which the IPI sampled in a manner analogous to the experiments conducted to investigate external and internal OH removal discussed above. Ozone mixing ratios in the range 0–2.5 ppmv were generated by varying the Hg lamp current (0–21 mA), while water vapour levels in the range 0.1–1.0% were produced by varying the flow through the bubbler, or by bypassing it completely, and the total flow (32 slm) was

compensated by changing the dry zero air flow. Laser power (LP) at 308 nm was varied in the range 3–17 mW by varying the ratio of acetone:water in a cuvette placed before the fibre launcher that is used to send laser light to the detection cells.

Figure 7 shows the results of $O_3$ + $H_2O$ vapour interference tests. It can be seen that the interference signal ($OH_{int}$ = $OH_{wave}$ − $OH_{chem}$) is linear in both ozone (panel (a)) and water vapour (panel (b)) mixing ratios.

The quadratic dependence of the interference signal on laser power (panel (c)) in terms of raw count rates indicates that the interference originates from a two-photon process, as expected. However, since OH data are normalised to laser power, the equivalent OH concentrations are linear with respect to laser power. Thus overall, $OH_{int}$ is linear in ozone, water vapour, and laser power. Normalisation of the slope in panel (a) yields the relation:


$$[OH_{int}] = (520 \pm 140) \text{ molecule cm}^{-3} \text{ ppbv}^{-1} \text{ \%}^{-1} \text{ mW}^{-1} \times [O_3] \times [H_2O] \times LP \tag{7}$$

where [$O_3$], [$H_2O$], and LP are in units of ppbv, %, and mW, respectively. Under typical atmospheric conditions of [$O_3$] = 50 ppbv and [$H_2O$] = 1%, and a typical instrument laser power of 15 mW, the interference

signal is equivalent to an OH concentration of $3.9 \times 10^5$ molecule cm$^{-3}$. This signal is slightly smaller than the instrumental limit of detection (LOD) of ~$7 \times 10^5$ molecule cm$^{-3}$ at a signal-to-noise ratio (SNR) of 2,



but nonetheless it was used to correct the ambient OHwave data presented in section 3.3, using co-located measurements of ozone and water vapour.

### 3.2.2 Isoprene ozonolysis

To test for interferences from isoprene (ISO) ozonolysis products, isoprene (~16 ppmv) and ozone (~1.8 ppmv) were mixed in the calibration wand and the scavenger (propane, PROP) was injected into the IPI flow tube. The propane concentrations were set to those used for ambient OHchem measurements, such that the tests were representative of normal atmospheric sampling (i.e., to test whether an interference signal would remain in ambient data). However, to generate sufficient OH signal for quantitative analysis, ozone and

isoprene were introduced at concentrations that far exceeded their typical ambient levels (Table 3). Unlike previous tests of interferences from alkene ozonolysis (Novelli et al., 2014b), low [$O_3$]:[ISO] ratios were used to suppress the signal contribution from the atmospheric (real) OH generated by ozonolysis (i.e., isoprene acted as an additional OH scavenger). To allow sufficient time for steady-state conditions to develop, the IPI did not sample from the calibration wand directly, but instead a 30 cm flow tube (ID ~ 19 mm) was used to

extend the IPI (which sampled wand gas at the normal IPI flow rate of ~32 slm, $\tau$ ~ 0.15 s).

Time series of the interference testing experiments conducted using the IPI are shown in Figure 8. In panel (a), no isoprene is added, but due to ozone photolysis in the presence of high [$H_2O$] (0.73%) an interference signal (OH$_{int}$) is observed (i.e., signal in the presence of propane is higher than the offline signal). The magnitude of this signal (OHint ~ $1.0 \times 10^7$ molecule cm$^{-3}$) yields a scale factor of $510 \pm 270$ molecule cm$^{-3}$

ppbv$^{-1}$ %$^{-1}$ mW$^{-1}$ when linearly extrapolated down from the measured [$O_3$], [$H_2O$], and LP, in agreement with the $520 \pm 140$ molecule cm$^{-3}$ ppbv$^{-1}$ %$^{-1}$ mW$^{-1}$ in equation (E7).

In panel (b), ozone and isoprene react under dry conditions, and an interference signal is observed again. The low $H_2O$ (0.07%) suppressed the $O_3$ + $H_2O$ interference, such that this cannot explain the magnitude of OHint (~$1.9 \times 10^7$ molecule cm$^{-3}$, Table 3), suggesting that OH was formed internally from a reaction other

than $O^1D$ + $H_2O$. Under high-humidity ($H_2O$ ~ 1%) conditions (panel (c)), OHint (~$1.6 \times 10^7$ molecule cm$^{-3}$) was similar, but in this case the signal can be explained almost entirely by the $O_3$ + $H_2O$ interference. Under dry conditions but with a ten-fold higher concentration of propane (as used for the AIRPRO summer campaign, panel (c)), the interference signal from panel (b) was reduced but remained elevated relative to the offline signal (OHint ~ $1.4 \times 10^7$ molecule cm$^{-3}$), where again the contribution from $O_3$ + $H_2O$ cannot explain

the discrepancy. The decrease in OHint between panels (b) and (d) may be attributed to the suppression of steady-state OH generated from ozonolysis, but the remaining signal implies that OH was also formed internally in both cases. For the dry, low-propane experiment (panel (b)), the magnitude of the OH signal is much higher than that calculated from a steady-state model (~$1.4 \times 10^6$ molecule cm$^{-3}$).

The suppression of the interference signal attributable to $O_3$/isoprene only (i.e., $O_3$ + $H_2O$ corrected) by

the addition of water vapour (panel (c), $H_2O$ ~ 1%) suggests that the internal OH may have been formed from SCIs. The simplest C1 and C2 SCIs are known to react quickly with the water vapour dimer ($k$ ~ $4$–$7 \times 10^{-12}$ cm$^3$ molecule$^{-1}$ s$^{-1}$ at 298 K for $CH_2OO$ (Chao et al., 2015; Lewis et al., 2015)) and monomer ($k$ ~ $1$–$2 \times$



$10^{-14}$ cm$^3$ molecule$^{-1}$ s$^{-1}$ for *anti*-CH$_3$CHOO (Taatjes et al., 2013; Sheps et al., 2014; Lin et al., 2016)), respectively. Reaction with the water vapour monomer was also shown to be relatively fast ($k \sim 1.2 \times 10^{-15}$

cm$^3$ molecule$^{-1}$ s$^{-1}$, $k_{loss} \sim 300$ s$^{-1}$ at ~1% H$_2$O) for the ensemble of SCIs, including the C1 SCI, generated from isoprene ozonolysis (Newland et al., 2015).

However, regardless of whether the signal observed at high propane is due to internally formed OH, which may have originated from SCIs, the equivalent OH concentrations are negligible when extrapolated back to ambient chemical conditions (Table 3). Assuming a linear dependence of the interference signal on both

ozone and isoprene, the interference (after O$_3$ + H$_2$O correction) is equivalent to <$10^2$ molecule cm$^{-3}$ at the ozone (10 ppbv) and isoprene (3.5 ppbv) levels measured in a low-NO$_x$, biogenic environment during the Oxidants and Particle photochemical processes (OP3) campaign in Borneo, 2008 (Hewitt et al., 2010). Similarly, the interference is calculated to be higher for the ozone (90 ppbv) and isoprene (7.9 ppbv) levels during the summer AIRPRO campaign, but it is still <$10^3$ molecule cm$^{-3}$. The insignificance of the

interference signal for atmospherically relevant O$_3$/alkene concentrations is consistent with the results of previous interference experiments, for which equivalent OH concentrations of ~3–4 $\times 10^4$ (Novelli et al., 2014b; Fuchs et al., 2016) and ~4 $\times 10^5$ molecule cm$^{-3}$ (Rickly and Stevens, 2018) can be derived.

### 3.2.3 NO$_3$ radicals

Fuchs et al. (2016) found that, despite the absence of a hydrogen atom, NO$_3$ radicals were responsible for a

small OH interference signal in the Forschungszentrum Jülich (FZJ) LIF–FAGE instrument, equivalent to an OH concentration of $1.1 \times 10^5$ molecule cm$^{-3}$ per 10 pptv NO$_3$. The OH interference scaled linearly with observed NO$_3$ mixing ratios but showed no dependence on inlet length, cell pressure, laser power, or humidity, and the background signal did not change significantly in the presence of CO scavenger, suggesting the OH was indeed being formed internally. It was postulated that the interference was a result of a

heterogeneous process involving NO$_3$ and H$_2$O adsorbed on instrument walls. Interference signals were also observed in the detection of HO$_2$ and RO$_2$ radicals, equivalent to $1.0 \times 10^7$ and $1.7 \times 10^7$ molecule cm$^{-3}$, respectively, per 10 pptv NO$_3$.

To test for an NO$_3$ interference in OH measurements made by the Leeds FAGE instrument, NO$_3$ was generated from the reaction of ozone and NO$_2$:


O$_3$ + NO$_2$ → NO$_3$                     (R1)

NO$_3$ + NO$_2$ → N$_2$O$_5$              (R2)

N$_2$O$_5$ → NO$_3$ + NO$_2$             (R3)

In these experiments, ozone was generated by flowing zero air (15 slm) past a Hg lamp (LOT LSP035). A constant 0.5 slm flow of NO$_2$ (BOC, 2 ppmv) was diluted in 25 slm zero air and mixed with the zero air/ozone flow as well as an additional zero air dilution flow of 10 slm, to yield a final mixing ratio of 20 ppbv. Gas was delivered to the IPI system using the calibration wand, with a total residence time of 3.7 s for the O$_3$ +





NO$_2$ reaction. Ozone mixing ratios in the range 0–2.8 ppmv (after dilution) were generated by varying the

current supplied to the Hg lamp. NO$_3$ radical mixing ratios in the range 0–90 pptv were calculated based on a box model with rate constants taken from the Master Chemical Mechanism (MCM; http://mcm.leeds.ac.uk/MCM) version 3.3.1 ($k_{R1} = 3.52 \times 10^{-17}$ cm$^3$ molecule$^{-1}$ s$^{-1}$, $k_{R2} = 1.24 \times 10^{-12}$ cm$^3$ molecule$^{-1}$ s$^{-1}$, and $k_{R3} = 0.045$ s$^{-1}$). These experiments were performed under dry conditions (H$_2$O ~ 0.07%), such that only a small correction was applied for the O$_3$/H$_2$O interference.

The results of the NO$_3$ radical interference tests are shown in Figure 9. It can be seen that the equivalent OH signals were all $< 8 \times 10^5$ molecule cm$^{-3}$, and almost always below the instrument LOD of $6.3 \times 10^5$ molecule cm$^{-3}$ (SNR = 2). Unlike the dependence found by Fuchs et al. (2016), the interference signal does not increase linearly with NO$_3$. However, based on the point at the highest NO$_3$ mixing ratio of ~90 pptv, the interference is equivalent to an insignificant $\sim 6 \times 10^4$ molecule cm$^{-3}$ at 10 pptv NO$_3$, or approximately half

of that observed by Fuchs et al. (2016). These experiments suggest that an interference from NO$_3$ radicals is not significant for the detection of OH using the Leeds ground-based FAGE instrument.

### 3.3 Ambient observations of OHwave and OHchem

### 3.3.1 ICOZA 2015

Figure 10 shows the overall intercomparison of OHwave (with O$_3$ + H$_2$O interference as given in Equation 7

subtracted) and OHchem measurements made during the ICOZA 2015 campaign. It is evident that the raw data (averaged for 4 minute periods) are quite noisy, but averaging to 1 h improves the precision and reveals a tight correlation, with the majority of points scattered around the line of 1:1 agreement. An orthogonal distance regression (ODR) fit (Boggs et al., 1987), which accounts for errors in both the $y$- and $x$- directions, to the hourly data yields a slope of $1.160 \pm 0.058$ ($2\sigma$) and a negative intercept on the order of the instrumental

precision. In a similar manner, an unweighted least squares linear fit (not shown) gives a slope of $1.060 \pm 0.065$, an intercept of $(0.5 \pm 1.5) \times 10^5$ molecule cm$^{-3}$, and a correlation coefficient ($R^2$) of 0.992. Regardless of the fit method, these results show that on average, the two OH measurements agree within the instrumental uncertainty of ~26% at $2\sigma$.

Median hourly diurnal profiles of OHwave, OHchem, and $J$(O$^1$D), averaged over both IPI sampling

periods, are shown in Figure 11. The two OH measurements exhibit virtually identical profiles, with peak values of $\sim 3 \times 10^6$ molecule cm$^{-3}$ slightly before solar noon, and relatively high concentrations ($\sim 1$–$2 \times 10^6$ molecule cm$^{-3}$) persisting into the early evening despite the concomitant falloff in $J$(O$^1$D). Nighttime levels were generally below $5 \times 10^5$ molecule cm$^{-3}$. The variability in OH concentrations, shown only for OHchem for clarity, was high during both day and nighttime periods.

In Figure 10 it can be seen that some points lie substantially above the 1:1 line, especially for the 4 minute averaged raw data. It is possible that, despite the good overall agreement between the median diurnal profiles of OHwave and OHchem in Figure 11, OHint may have exhibited its own distinct diurnal profile, independent of atmospheric OH concentrations, for example if the interference signal was generated from a particular chemical species. However, the median diurnal profile of individual OHint measurements (= OHwave –





OHchem) in Figure 11 exhibits no obvious structure, with values scattered around zero and a mean $\pm 2\sigma$ value of $(0.3 \pm 3.3) \times 10^5$ molecule cm$^{-3}$, which is well below the LOD for individual OH measurements. Similarly, the average (OHwave − OHchem)/OHwave ratio (i.e., the contribution of interferences to the total OHwave signal) was zero within error (mean $\pm 2$ standard errors (SE) = 0.03 $\pm$ 0.12).

    Furthermore, OHwave and OHchem exhibit virtually identical behaviour when binned against various

parameters (Woodward-Massey, 2018), including those previously implicated in LIF–FAGE measurement interferences, such as $J(O^1D)$ (Feiner et al., 2016), temperature (Mao et al., 2012; Novelli et al., 2017), OH reactivity $k'_{OH}$ (Mao et al., 2012), OH reactivity due to VOCs, and $O_3$ (Feiner et al., 2016; Novelli et al., 2017) and NO (Feiner et al., 2016) mixing ratios.

### 3.3.2 AIRPRO winter 2016

The overall agreement between the two measurements is presented in the correlation plot in Figure 12. As with ICOZA (Figure 10), a tight correlation is revealed after averaging the data to one hour, and all points are distributed evenly around the line of 1:1 agreement. ODR fitting yields an overall slope of $1.051 \pm 0.039$ and a negative intercept of a similar magnitude to the instrumental precision. An unweighted least squares linear fit (not shown) gives a slope of $0.997 \pm 0.038$, an intercept of $(5.1 \pm 7.3) \times 10^4$ molecule cm$^{-3}$, and an

$R^2$ of 0.97.

    The two measurements exhibit the same profile on a diurnal basis (Figure 13), with a diel maximum of $\sim 3 \times 10^6$ molecule cm$^{-3}$ occurring in the late morning due to the build-up of HONO overnight. At night, OHchem concentrations were close to the LOD ($< \sim 2 \times 10^5$ molecule cm$^{-3}$), while OHwave measurements were frequently negative, possibly as a result of over-subtraction of the $O_3/H_2O$ interference as this is subject to

high uncertainty (Figure 7). The diurnal profile of OHint is scattered around zero with a mean $\pm 2\sigma$ difference of $(-0.9 \pm 2.7) \times 10^5$ molecule cm$^{-3}$, and the mean $\pm 2$ SE contribution of interferences to the total signal was $-0.02 \pm 0.07$.

### 3.3.3 AIRPRO summer 2017

    The intercomparison of OHwave and OHchem measurements for the AIRPRO summer campaign is shown

in Figure 14. Consistent with ICOZA and the AIRPRO winter results, the 1 h data are scattered around the 1:1 line, with an overall ODR slope of $1.103 \pm 0.017$. However, the intercept is more negative than for the other campaigns, which suggests that the $O_3/H_2O$ interference may have been overestimated, as it is during this campaign that the highest ozone mixing ratios (~90 ppbv diurnally-averaged maximum, Table 1) were encountered. Similarly, an unweighted least squares linear fit to the data (not shown) yields a slope of 1.111

$\pm 0.029$, an intercept of $(-3.8 \pm 1.7) \times 10^5$ molecule cm$^{-3}$, and an $R^2$ of 0.92 (data not shown).

    Again, the two measurements follow the same diurnal profile (Figure 15), peaking in the afternoon at ~1 $\times 10^7$ molecule cm$^{-3}$ with relatively high nighttime levels of $\sim 1$–$2 \times 10^6$ molecule cm$^{-3}$. As with ICOZA and the AIRPRO winter campaign, the OHint diurnal profile does not exhibit any obvious structure, with values scattered around zero and a mean $\pm 2\sigma$ difference of $(-1.6 \pm 4.1) \times 10^5$ molecule cm$^{-3}$. The mean $\pm 2$ SE


(OHwave – OHchem)/OHwave ratio was −0.09 ± 0.10. During AIRPRO, measured $NO_3$ mixing ratios reached up to ~100 pptv, such that the lack of significant nighttime OH interference signals is consistent with the results of $NO_3$ interference tests (section 3.2.3).

     It can be seen from Figure 14 that there is a small cluster of points that lie significantly away from both the 1:1 and ODR regression lines, which are characterised by high OHwave concentrations of $> 1.5 \times 10^7$

molecule $cm^{-3}$. This finding was investigated further, with the results summarised in Figure 16. Above an OHchem threshold of ~1.0–1.5 × $10^7$ molecule $cm^{-3}$, the OHint signal becomes significantly greater than zero and the instrument LOD, reaching ~3–5 × $10^6$ molecule $cm^{-3}$ at OHchem levels of around ~2 × $10^7$ molecule $cm^{-3}$. However, these results should be treated with caution, since only a few points are available for which OHchem was present at such high concentrations. The same behaviour was not observed for either

the AIRPRO winter or ICOZA campaigns, since OHchem levels did not surpass $1 \times 10^7$ molecule $cm^{-3}$, but the analogous mean values at low OHchem concentrations are in agreement with the AIRPRO summer results.

     The above results suggest that in the Beijing summertime, the Leeds FAGE instrument is subject to an interference(s) at the highest OH levels, although its contribution of ~15–20% (Figure 16) is still below the

instrumental accuracy of 26% at $2\sigma$. This finding is consistent with the suggestion of (Fittschen et al., 2019) that ROOOH species, formed from $RO_2$ + OH reactions, generate an OH interference in LIF–FAGE instruments, since high OH levels would generate high $RO_2$ concentrations and favour this class of reaction. It is also possible that, for high ambient OH production rates, the scavenger cannot react with the sampled OH sufficiently quickly, leading to elevated but spurious OHchem background signals. Although a modelling

study of the inlet chemistry would be required to fully assess this hypothesis, it is likely not the case considering that the propane concentration used during AIRPRO results in an OH lifetime of ~0.03 ms in the IPI flow tube, in comparison to a residence time of ~20 ms (i.e., ~700 OH lifetimes).

## 4 Discussion

The results from the three field campaigns that feature in this work demonstrate that the Leeds ground-based

FAGE instrument does not suffer from substantial interferences in the measurement of OH using the conventional, wavelength-modulation background technique, OHwave. This is illustrated best by the slopes of the overall measurement intercomparison plots (Figures 10, 12, and 14), which ranged from 1.05–1.16, well within the instrumental uncertainty of ~26% at $2\sigma$.

     With respect to previous studies during which OH has been measured by a LIF instrument equipped with

a scavenger injector, the significance of interferences during the campaigns that feature in this work are amongst the lowest observed (Table 4). This can likely be attributed to two main factors: environment and instrumental. In terms of the former, none of the studies described in the present study took place in forested environments, where the most significant interferences have been observed (Mao et al., 2012; Novelli et al., 2014a; Feiner et al., 2016). However, as mentioned previously, the AIRPRO summer campaign did share

some characteristics, in that high BVOC and low NO mixing ratios were observed in the afternoon. Despite

this, OHwave and OHchem were in good agreement. These findings provide confidence in previous measurements of OH using the same instrument, particularly those in a forested region (Whalley et al., 2011), and support the hypothesis that there are unknown OH sources in this type of environment.

The insignificance of daytime interferences during the AIRPRO campaigns are consistent with results of another urban study, CalNex-LA (Research in California at the Nexus of Air Quality and Climate Change) (Griffith et al., 2016). The $O_3/H_2O$ interference is much higher (up to ~$4 \times 10^6$ molecule cm$^{-3}$ OH equivalent during CalNex-LA) in the Indiana University (IU) LIF instrument (Dusanter et al., 2009), such that the daytime contributions of ~33% can be explained entirely by this known interference. However, measurements made at a nearby site during the same study (CalNex-SJV) showed daytime contributions of ~20% (Brune et al., 2016), although this may be related to instrumental differences as discussed below. On average, interferences were not observed in the daytime during ICOZA, but they were observed in other coastal campaigns, namely DOMINO (Diel Oxidants Mechanisms In relation to Nitrogen Oxide) HO$_x$ (~50%) (Novelli et al., 2014a) and CYPHEX (CYprus PHotochemistry EXperiment, ~45%) (Mallik et al., 2018), as well as in rural regions, HOPE (Hohenpeißenberg Photochemistry Experiment, 20–40%) (Novelli et al., 2014a). Studies in the North China Plain have revealed small interferences on the order of 0–10% (Tan et al., 2017; Tan et al., 2018), with slightly higher but variable contributions of 0–20% in the Pearl River Delta (Tan et al., 2019).

The second major reason for the differences in contributions between the studies listed in Table 4 is likely instrumental effects. For the campaigns in which the highest OH interferences have been observed (Mao et al., 2012; Novelli et al., 2014b; Feiner et al., 2016), OH measurements were made using the Max Planck Institute (MPI) (Martinez et al., 2010) and PSU (Faloona et al., 2004) LIF instruments. These instruments feature laser-multi-pass detection cells, which give rise to larger detection volumes and increased UV fluence, although this may not be relevant considering that the interference signals did not display any laser power dependence for these instruments. The Leeds instrument also differs in terms of cell geometry, where the HO$_x$ cell is composed of a short (5 cm) turreted inlet on top of a large fluorescence cell (additional ~8 cm to laser axis, ~13 cm total length, and a cell diameter of 25 cm). In contrast, the MPI and PSU instruments feature flow tube-like inlets (14–17 cm from the pinhole to laser axis) mounted on smaller fluorescence cells, facilitating the interaction of sampled gas with the cell walls, which may promote the generation of internal OH. For the measurements listed in Table 4, the Peking University (PKU) instrument (Tan et al., 2017) is most similar to the Leeds FAGE (i.e., single-pass detection, ~10 cm total length from sampling inlet to laser axis), for which similar daytime interferences on the order of ~0–20% were observed.

For the ICOZA campaign, nothing could be inferred about the origin of the OH interference signal when one was observed, as it did not exhibit any characteristic diurnal profile (Figure 11), and showed no obvious dependences on a variety of meteorological and chemical parameters. This finding is in contrast to previous studies in which diel profiles (Mao et al., 2012; Feiner et al., 2016) and dependences (Mao et al., 2012; Feiner et al., 2016; Novelli et al., 2017) of the interference have been observed. The occurrence of large (i.e., $> 1 \times 10^6$ molecule cm$^{-3}$) background OH signals (OHint = OHwave − OHchem) after instrumental problems (e.g.,



power cuts, data not shown) implies that the differences may have been instrumental rather than as a result of a species present in ambient air, although the data at these times did pass all quality control filters and

therefore could not be rejected. Nonetheless, any differences are still a concern, regardless of their cause; the IPI system thus serves as an additional check on measurement accuracy and operational stability, and is perhaps most useful for fieldwork sites where power supplies are unreliable, for example in more remote areas.

It is possible that, even though the background OH had a flat diurnal profile in each field campaign, the

species responsible for any interference observed were different between day and nighttime periods. Thus, analysis of the day and nighttime data separately, as a function of the same parameters, might reveal more information. Considering the recent identification of $NO_3$ radicals as an internal OH source in LIF instruments (Fuchs et al., 2016), and that OH concentrations have often been underpredicted at night (Faloona et al., 2001; Mao et al., 2012; Ren et al., 2013; Hens et al., 2014; Lu et al., 2014; Tan et al., 2017), this is perhaps the most

interesting period for further study. However, for the data presented in this work, robust quantitative nighttime analyses are not possible due to OH measurements being below or close to the instrument LOD.

In this work, there are several key findings that stand out. First, OHwave and OHchem were in good agreement even at the very low NO concentrations of < 100 pptv during ICOZA, and the moderate afternoon levels (~500 pptv on average but often < 100 pptv (Shi et al., 2018)) during the AIRPRO summer campaign.

While the role of isoprene could not be assessed for ICOZA, due to the limited range of concentrations observed (< 0.2 ppbv), it reached high levels during AIRPRO summer (up to 7.9 ppbv, larger than seen in some forested regions) but did not seem to perturb the agreement between the two measurements. In addition, very high levels of aromatic VOCs were observed during both AIRPRO winter and summer, where the agreement between OHwave and OHchem suggests that the intermediates of aromatic oxidation, such as

exotic bicyclic species (Birdsall et al., 2010) and highly-oxygenated molecules (HOMs) (Wang et al., 2017; Molteni et al., 2018; Hammes et al., 2019), do not give rise to OH interferences, which is postulated to be the case for intermediates (SCIs) in the ozone-oxidation of alkenes (Novelli et al., 2014b; Novelli et al., 2017; Rickly and Stevens, 2018). However, the large alkene and ozone concentrations observed during AIRPRO summer should favour the formation of these SCIs, but significant interferences were not observed, consistent

with laboratory investigations of the isoprene interference and casting doubt on the SCI hypothesis. Although, the AIRPRO SCI concentrations also depend on the magnitude of the SCI loss rates, which could be high if elevated levels of $SO_2$ (Welz et al., 2012; Sheps et al., 2014) or organic acids (Welz et al., 2014) were present.

Considering the success of the first three field deployments of the IPI system, and given that it does not reduce the instrument sensitivity towards OH, it is suggested that the system is adopted for permanent use in

ambient studies, although conventional sampling should still be performed from time-to-time to check for potential artefacts caused by the IPI system itself. Another advantage of the IPI system is that it reduces the amount of solar light entering the pinhole, which reduces the size and variability of daytime background signals and therefore improves signal-to-noise and hence detection limits. It is recommended that the IPI propane concentration is kept the same as the summer AIRPRO campaign, as it is possible that the slightly



poorer agreement between OHwave and OHchem during ICOZA was because of the lower propane flow used (i.e., the flow was not sufficient to ensure that OH generated from all steady-state sources was removed), although this cannot be verified.

Future field campaigns using the IPI will allow for the assessment of interferences in the Leeds FAGE instrument for a range of different environments. From these, the contribution of interferences for previous

studies in similar environments, where measurements were made prior to the discovery of significant interferences in the LIF measurement of OH reported by others, may be inferred. The measurement-model comparisons may then be reassessed in light of any new information regarding the accuracy of OH measurements. Regardless of the reasons for any differences between the two measures of OH (i.e., chemical interferences or instrumental problems such as during recovery periods after power cuts), the IPI system

serves as an additional check on OH observations, increasing confidence in the validity of the data obtained.

**5 Conclusions**

The addition of an IPI system to the Leeds ground-based FAGE instrument allowed for a comprehensive investigation of OH measurement interferences in both the laboratory and the field. Following its optimisation and thorough characterisation in terms of sensitivity and external and internal OH removal

efficiency, laboratory experiments were conducted to assess potential interferences from (1) the photolysis of $O_3$ in the presence of $H_2O$ vapour, (2) the intermediates and products of isoprene ozonolysis, and (3) $NO_3$ radicals. For $O_3 + H_2O$, a small but potentially significant interference (at high $O_3$ levels) was found, but interferences from isoprene ozonolysis products and $NO_3$ radicals were shown to be insignificant under typical atmospheric conditions.

Field campaigns conducted in the UK and China showed that, on average, the Leeds ground-based FAGE instrument does not suffer from significant interferences in the detection of OH. It was only under the very high OH levels of $> 1.5 \times 10^7$ molecule $cm^{-3}$ sometimes observed during the AIRPRO summer campaign that interferences were found consistently, although their contributions (~15–20%) were smaller than the instrumental accuracy of 26% at $2\sigma$. Large interference signals ($> 1 \times 10^6$ molecule $cm^{-3}$) were occasionally

observed during the ICOZA campaign, but always after instrumental problems such as power cuts, suggesting that the OHchem method serves as an additional tool for verifying instrument stability and validating measurements. The Leeds IPI system will find continued use in future fieldwork.

**Acknowledgements**

We are grateful to the Natural Environment Research Council for funding (grant numbers NE/K012029/1, NE/K012169/1, NE/N007115/1 and NE/N006895/1). RWM, DRC and EJS are grateful to NERC for funding PhD studentships. We would like to acknowledge Brian Bandy, Grant Forster, David Oram and Claire Reeves (University of East Anglia), William Bloss, Leigh Crilley and Louisa Kramer (University of Birmingham), and Rachel Dunmore, Jacqui Hamilton, James Hopkins, James Lee, Chris Reed and Freya Squires (University

of York) for provision of some of the data used to generate the averages in Table 1. We would also like to



thank other participants in the ICOZA and AIRPRO field campaigns. We are grateful for technical support from the University of Leeds mechanical and electronics workshops,

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



# Tables


**Table 1.** Overall meteorological and chemical conditions encountered during each field campaign, including example VOCs. Values are given as the median of all points that coincide with IPI sampling periods, except for $J(O^1D)$ and $O_3$, which are reported as diurnally-averaged maxima.

|  | **ICOZA** | **AIRPRO winter** | **AIRPRO summer** |
|---|---|---|---|
| *Dates (IPI sampling)* | 3–8 and 12–16 Jul, 2015 | 2–7 Dec, 2016 | 23 May–25 Jun, 2017 |
| *Location* | 52º57'02''N, 1º07'19''E | 39°58'28''N, 116°22'16''E | |
| *Meteorological* |  |  |  |
| Temperature (°C) | 16 | 6.1 | 26 |
| $H_2O$ (%) | 1.5 | 0.4 | 1.6 |
| Wind speed ($ms^{-1}$) | 5.8 | 0.9 | 0.4 |
| $J(O^1D)$ ($10^{-6}$ $s^{-1}$) | 16 | 3.5 | 19 |
| *Chemical* |  |  |  |
| $O_3$ (ppbv) | 42 | 15 | 90 |
| NO (ppbv) | 0.19 | 22 | 0.81 |
| $NO_2$ (ppbv) | 2.2 | 33 | 17 |
| CO (ppbv) | 100 | 1120 | 460 |
| Propane (ppbv) | 0.26 | 6.2 | 3.8 |
| Isoprene (ppbv) | 0.02 | 0.07 | 0.38 |
| Benzene (ppbv) | 0.03 | 1.4 | 0.46 |
| $k'_{OH}$ ($s^{-1}$) | 4.4 | 38 | 25 |

[a]Integrated Chemistry of OZone in the Atmosphere

[b]an integrated study of AIR pollution PROcesses in Beijing (Shi et al., 2019)





**Table 2.** Internal removal of OH (%, $\pm 2\sigma$) as a function of propane mixing ratio in the IPI flow tube, determined as shown in Figure 6 (see text for details).

| Propane (ppmv) | Experiment no. | Internal removal (%) |
|---|---|---|
| 110 | 1 | $-0.1 \pm 4.8$ |
| (used for ICOZA and AIRPRO winter) | 2 | $0.3 \pm 7.7$ |
| | 3 | $-0.9 \pm 16$ |
| | *Average ± 2 SD* | $-0.2 \pm 1.1$ |
| 550 | 1 | $1.0 \pm 9.6$ |
| 1100 | 1 | $1.9 \pm 12$ |
| (used for AIRPRO summer) | 2 | $4.2 \pm 11$ |
| | 3 | $2.5 \pm 11$ |
| | *Average ± 2 SD* | $2.8 \pm 2.3$ |


**Table 3.** Summary of interference tests with $O_3$ and isoprene (ISO) in the presence of propane (PROP), based on the data in Figure 8.

| Test | $O_3$ (ppmv) | $H_2O$ (%) | ISO (ppmv) | PROP (ppmv) | OHint (molecule $cm^{-3}$) | | | |
|---|---|---|---|---|---|---|---|---|
| | | | | | *Obs.* | [a]$O_3/H_2O$ corr. | [b]*OP3 levels* | [c]*AIRPRO 2017 levels* |
| A | 1.64 | 0.73 | 0 | 110 | $1.0 \times 10^7$ | 0 | N/A | N/A |
| B | 1.86 | 0.07 | 16 | 110 | $1.9 \times 10^7$ | $1.8 \times 10^7$ | 21 | 430 |
| C | 1.83 | 0.98 | 16 | 110 | $1.6 \times 10^7$ | $8.0 \times 10^5$ | 1 | 19 |
| D | 1.85 | 0.07 | 16 | 1100 | $1.4 \times 10^7$ | $1.3 \times 10^7$ | 15 | 310 |

[a]Corrected using equation (E7)
[b]Oxidant and Particle photochemical processes field campaign in Borneo, 2008: average $O_3$ = 10 ppbv, ISO = 3.5 ppbv
[c]Diurnally-averaged maximum $O_3$ = 90 ppbv, overall maximum ISO = 7.9 ppbv

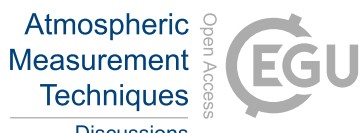

**Table 4.** Average contributions of FAGE background signals to the total OH measured ($= (OH_{wave} - OH_{chem})/OH_{wave}$) during ambient air studies where a chemical modulation technique was employed.

| Study | Year | Location | Environment Type | Contribution (%) Daytime | Contribution (%) Nighttime | Reference(s) |
|---|---|---|---|---|---|---|
| PROPHET | 1998 | N Michigan | Forest, isoprene dominated | Not tested | ~0 | Faloona et al (2001) |
| BEARPEX | 2009 | NE California | Forest, [a]MBO dominated | 40–60 | 50 | Mao et al (2012) |
| CABINEX | 2009 | N Michigan | Forest, isoprene dominated | Not tested | 50–100 | Griffith et al (2013) |
| SHARP | 2009 | Houston, Texas | Urban | 30 | 50 | Ren et al (2013) |
| CalNex-LA | 2010 | Pasadena, California | Urban, downwind of LA | 33* | Not reported | Griffith et al (2016) |
| CalNex-SJV | 2010 | Bakersfield, California | Urban | 20 | 80 | Brune et al (2016) |
| DOMINO | 2010 | El Arenosillo, near Huelva, SW Spain | Coastal, close to petrochemical industry | 50 | 100 | Novelli et al (2014) |
| HUMPPA-COPEC | 2010 | Hyytiälä, SW Finland | Boreal forest, terpene dominated | 60–80 | 100 | Hens et al (2014); Novelli et al (2014); Novelli et al (2017) |
| HOPE | 2012 | Hohenpeissenberg, S Germany | Rural | 20–40 | 100 | Novelli et al (2014); Novelli et al (2017) |
| SOAS | 2013 | near Brent, Alabama | Forest, isoprene dominated | 80 | >70 | Feiner et al (2016) |
| Wangdu | 2014 | North China Plain | Rural, urban influenced | 10 | Not reported | Fuchs et al (2017); Tan et al (2017) |
| CYPHEX | 2014 | NW Cyprus | Coastal, influenced by processed European emissions | 45 | 100 | Mallik et al (2018) |
| PRIDE-PRD2014 | 2014 | Pearl River Delta | Suburban, 60 km SW of Guangzhou | 0–20 | 0–20 | Tan et al (2019) |
| BEST-ONE | 2016 | North China Plain | Suburban, 60 km NE of Beijing | ~0 | ~0 | Tan et al (2018) |
| ICOZA | 2015 | N Norfolk Coast, UK | Coastal, London outflow | ~0 | ~0 | This work |
| AIRPRO Winter | 2016 | Beijing, China | Urban | ~0 | Nighttime OH almost always <LOD | This work |
| AIRPRO Summer | 2017 | Beijing, China | Urban | ~0 | ~0 | This work |

[a]MBO = 2-methyl-3-buten-2-ol, a biogenic volatile organic compound



# Figures

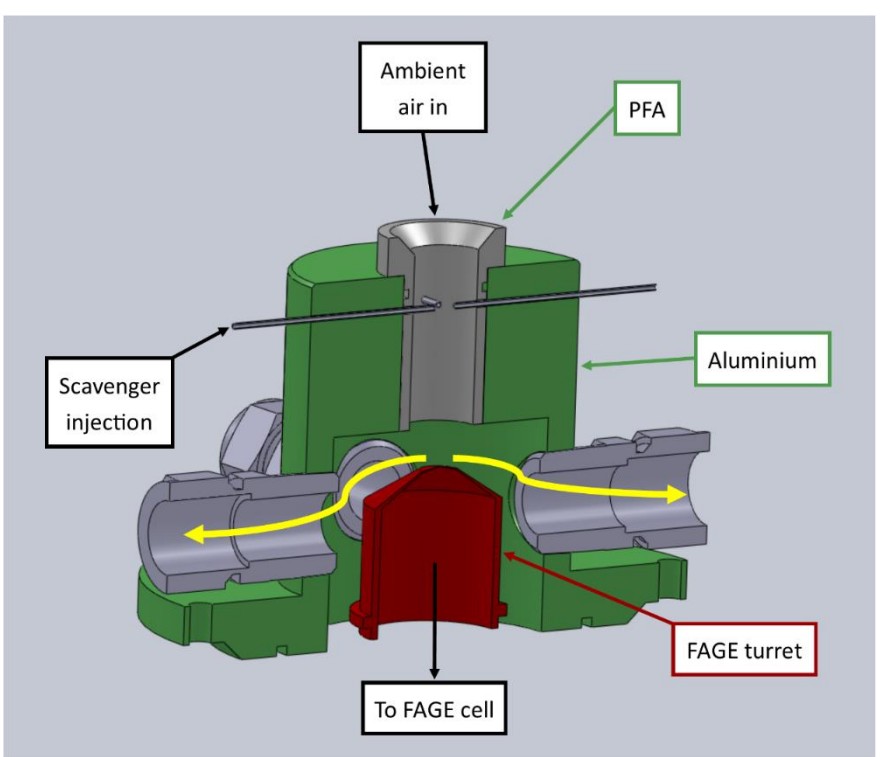

**Figure 1.** Labelled SolidWorks model of the Leeds inlet pre-injector (IPI). The scavenger is injected into the centre of the perfluoroalkoxy
5  (PFA) flow tube via four 0.25 mm ID needles. The thick yellow arrows indicate the direction of the sheath flow.





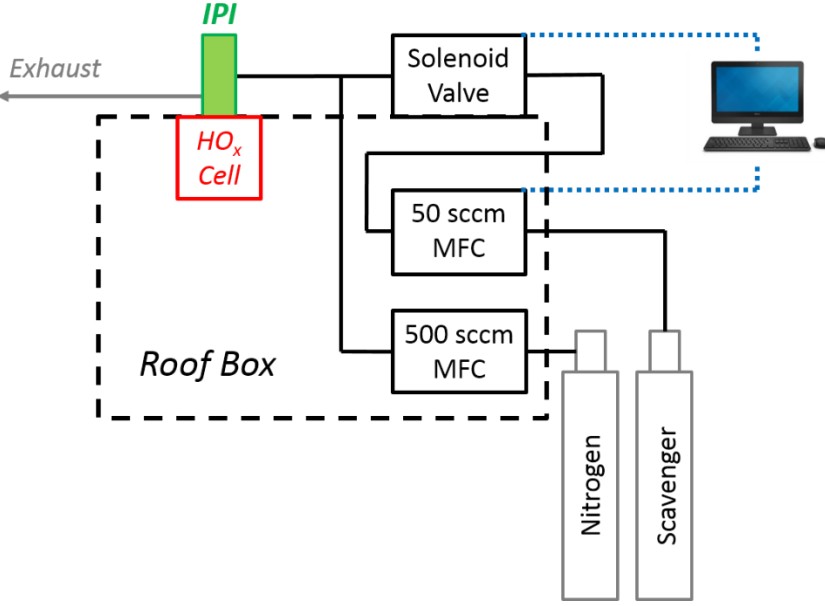

**Figure 2.** Diagram of the gas flows involved in IPI scavenger injection (not to scale). The two mass flow controllers (MFCs) are housed in the roof box, where the scavenger MFC (0–50 sccm) and injection valve (in a weatherproof housing on top of the roof box) are controlled using the main FAGE PC situated in the container laboratory.

**Figure 3.** Testing of OH losses in the IPI system. Each panel shows repeat measurements of OH signals ($\pm 2\sigma$) over the course of one day, where high OH concentrations were generated using a 184.9 nm Hg lamp placed near the instrument inlet. Blue and red markers denote individual measurements (one measurement "loop", i.e., one wavelength online–offline cycle) performed with ("IPIon") and without ("IPIoff") the IPI system, respectively. Solid lines correspond to the average signals for each day, with $2\sigma$ standard deviations (SD) shown by the dashed lines.



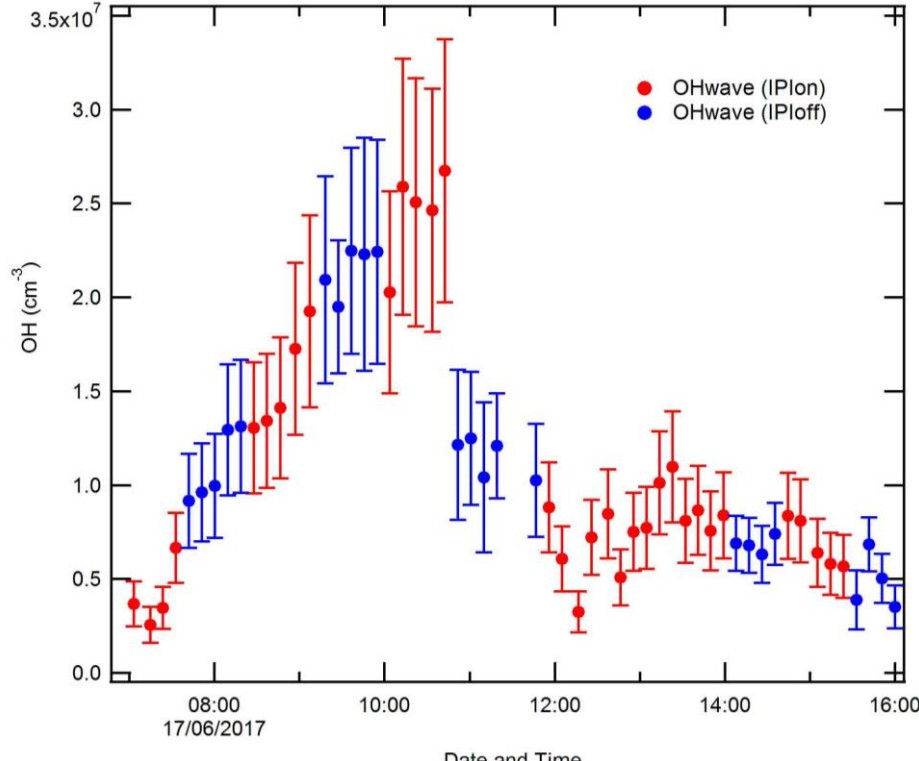

**Figure 4.** Time series of OHwave concentrations in Beijing on 17th June 2017, a period of high OH levels in the summer 2017 AIRPRO (an integrated study of AIR pollution PROcesses in Beijing) campaign. Blue and red markers (±2σ) denote observations made with and without the IPI system, respectively.





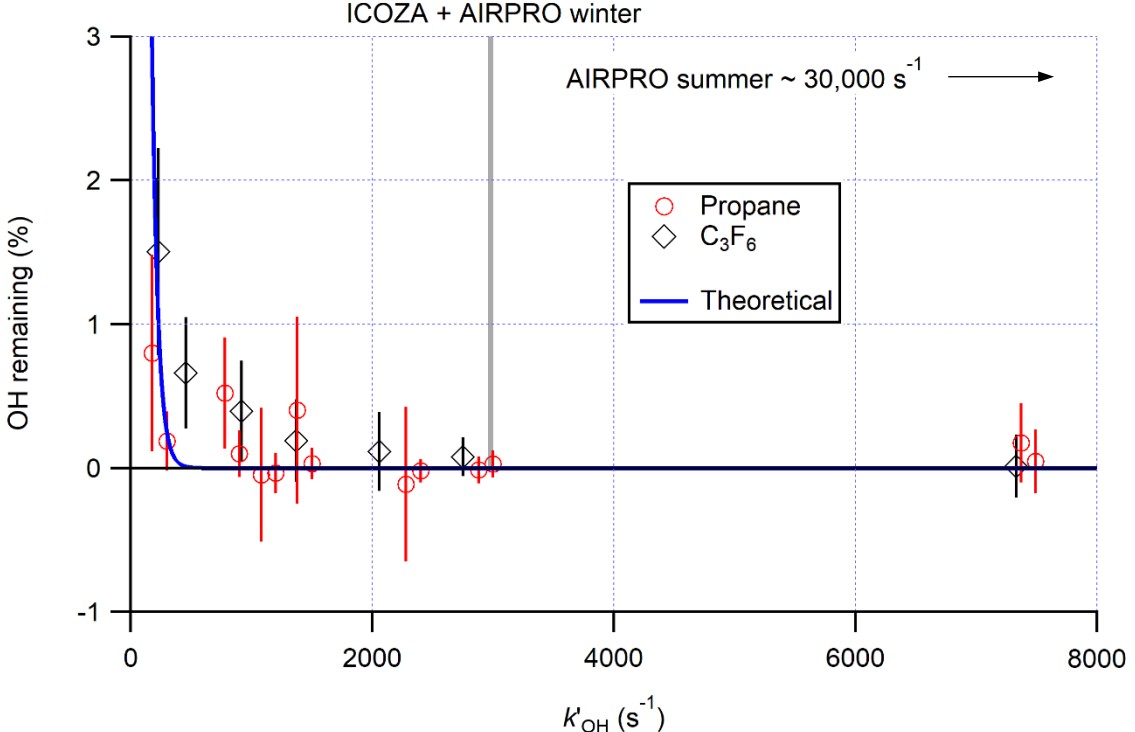

**Figure 5.** Proportion of the OH signal remaining (i.e., external OH removal efficiency) after addition of increasing concentrations of propane and perfluoropropene ($C_3F_6$) scavengers to the IPI flow tube, converted to equivalent OH reactivities ($k'_{OH}$) to account for the different rate constants for the reaction of each scavenger with OH. Error bars denote the $2\sigma$ SD of repeat experiments. The blue curve corresponds to the theoretical scavenging efficiency assuming perfect mixing, using the estimated residence time, $\tau \sim 20$ ms. The propane OH reactivity used for the ICOZA (Integrated Chemistry of OZone in the Atmosphere) and AIRPRO winter campaigns is given, but that used for AIRPRO summer is off-scale.





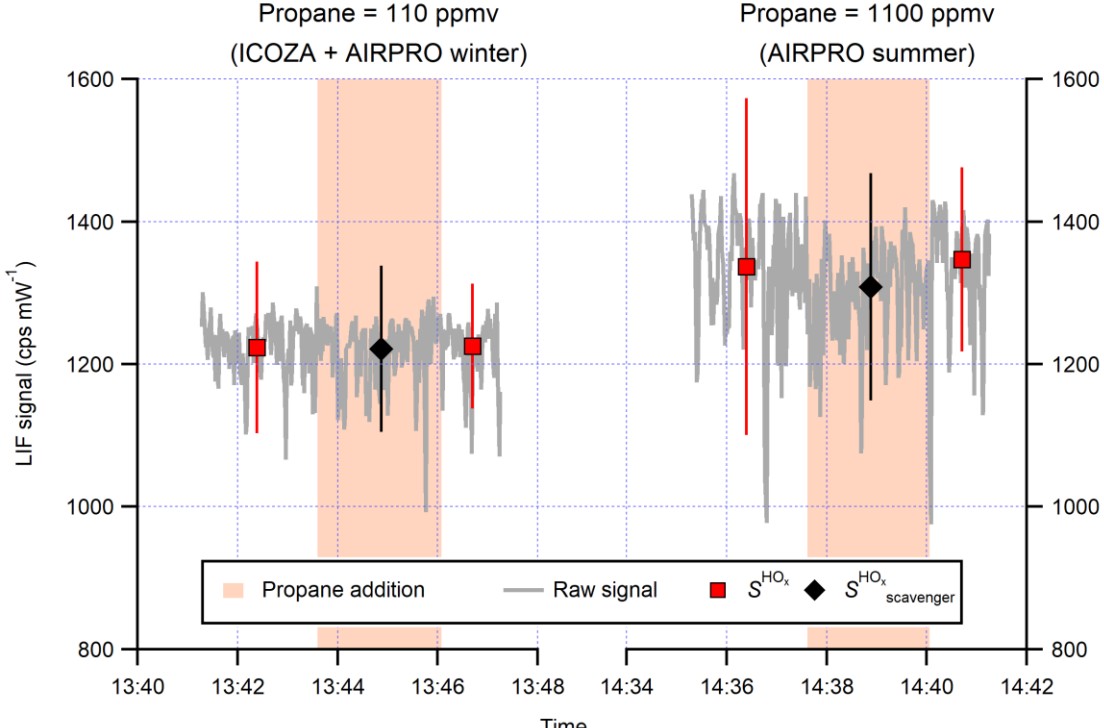

**Figure 6.** Time series of the LIF signal during internal OH removal experiments. The raw 1 s data are given by the grey line. NO was continuously added to the FAGE cell during these experiments, and points where propane was added to the IPI flow tube are indicated by the orange shaded panels, with the corresponding signal averages ($\pm2\sigma$) shown as markers (see text for details). The first experiment (left-hand side) corresponds to the propane mixing ratio used for ICOZA, while the second (right-hand side) corresponds to that used for AIRPRO. The results of the internal OH removal experiments are summarised in Table 2.





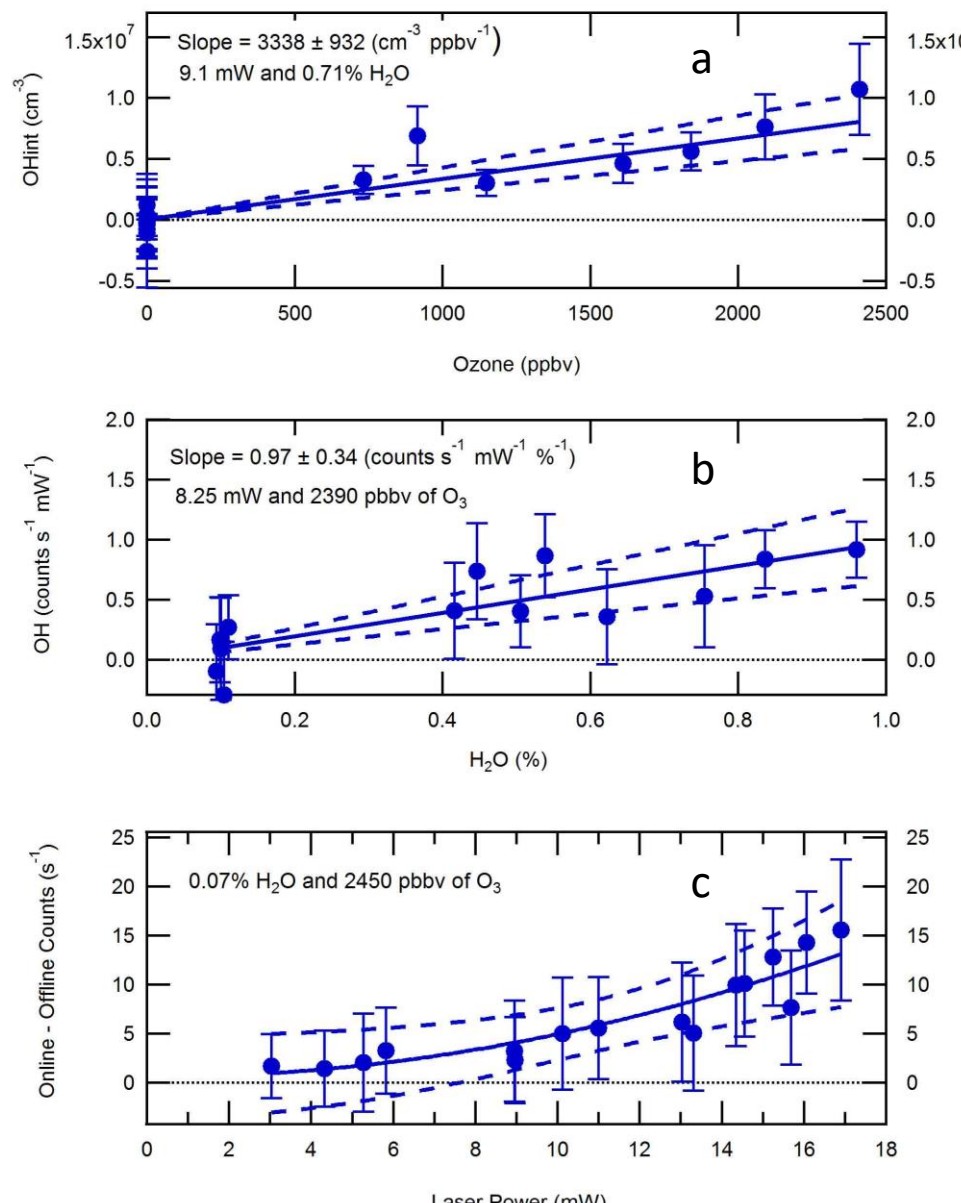

**Figure 7.** OH interference ($\pm 2\sigma$) from $O_3$ + $H_2O$ as a function of (a) $O_3$, (b) $H_2O$, and (c) laser power. The interference signal is linear in $O_3$ and $H_2O$ mixing ratios and quadratic in laser power, confirming the two-photon nature of the process. Normalising the slope in panel (a) to $O_3$ = 1 ppbv, $H_2O$ = 1%, and laser power = 1 mW yields an OH interference equivalent to a concentration of $520 \pm 140$ molecule cm$^{-3}$.

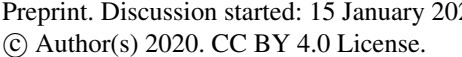

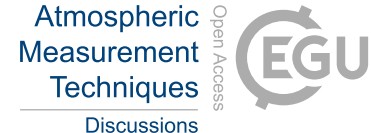

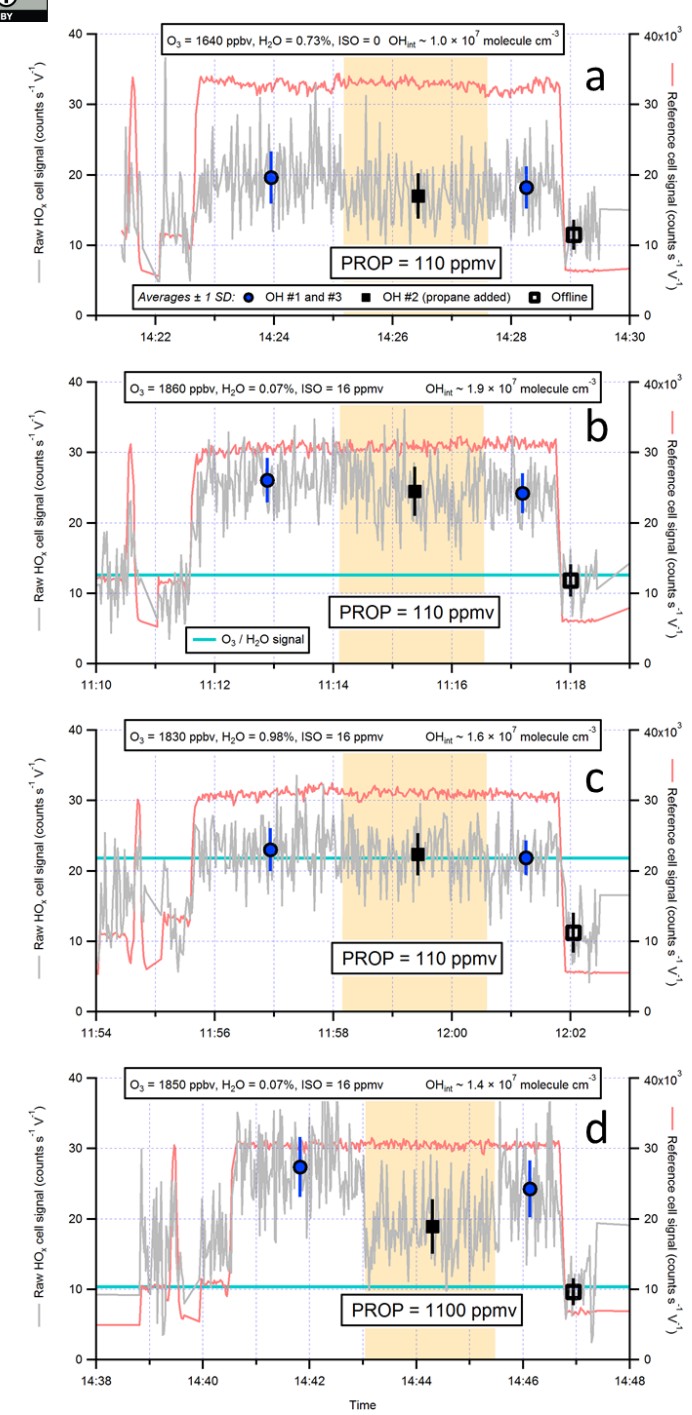

**Figure 8.** Isoprene (ISO) ozonolysis interference tests: (a) $O_3/H_2O$ only, (b) $O_3$ and ISO under dry conditions, (c) $O_3$ and isoprene with $H_2O$ added, and (d) $O_3$ and isoprene under dry conditions, but with a higher concentration of propane (PROP) to remove any steady-state generated OH. Shaded areas are periods of propane addition, and the light blue lines correspond to the calculated signals from $O_3 + H_2O$ only (for experiments with isoprene present). The interference signals ("OH #2" – "offline") were used to derive equivalent OH concentrations ($OH_{int}$), which are on the order of $\sim 1$–$2 \times 10^7$ molecule $cm^{-3}$. These experiments are summarised in Table 3. See text for further details.



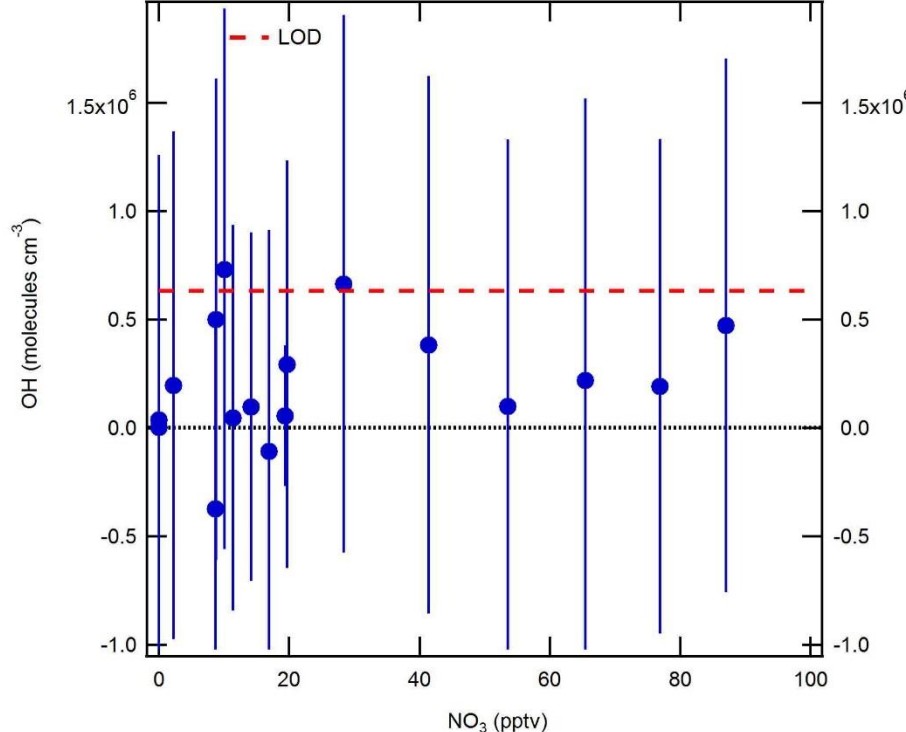

**Figure 9.** Equivalent OH concentrations (±2σ) measured during $NO_3$ radical interference tests. $NO_3$ concentrations were calculated using a box model and OH interference signals were corrected for the interference from $O_3$ + $H_2O$. The OH limit of detection (LOD, $6.3 \times 10^5$ molecule cm$^{-3}$, SNR = 2) is denoted by the red dashed line.



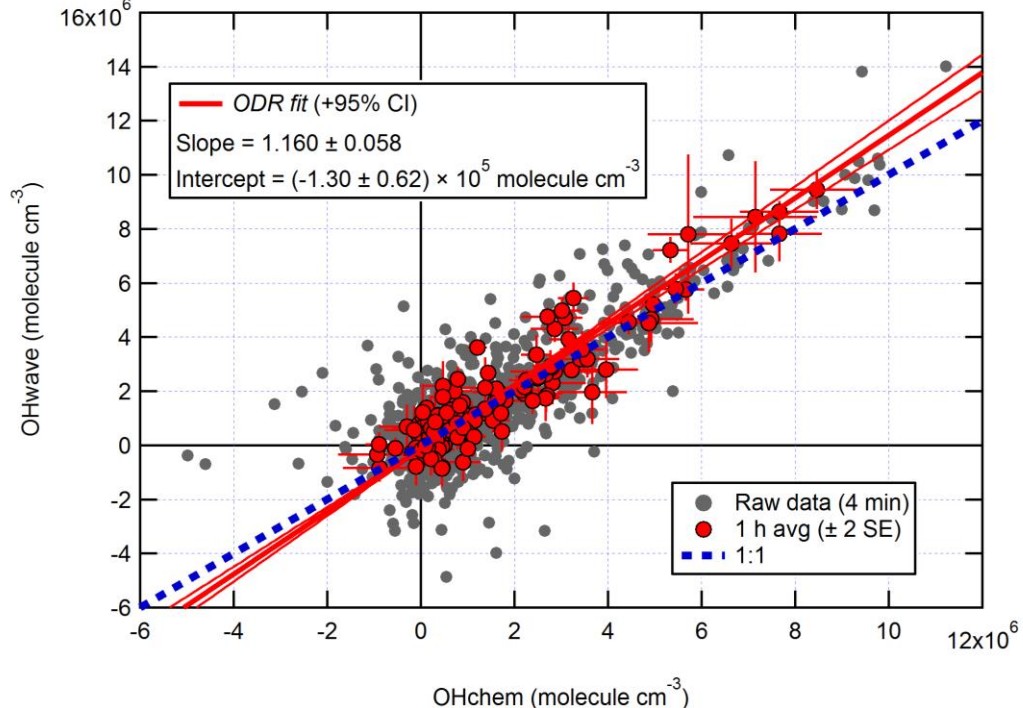

**Figure 10.** Overall intercomparison of OHwave and OHchem observations from the ICOZA campaign. Grey markers represent raw data (4 min), with 1 h averages (±2 standard errors (SE)) in red. The thick red line is the orthogonal distance regression (ODR) fit to the hourly data, with its 95% confidence interval (CI) bands given by the thin red lines; fit errors given at the $2\sigma$ level. For comparison, 1:1 agreement is denoted by the blue dashed line. OHwave data were corrected for the known interference from $O_3 + H_2O$.

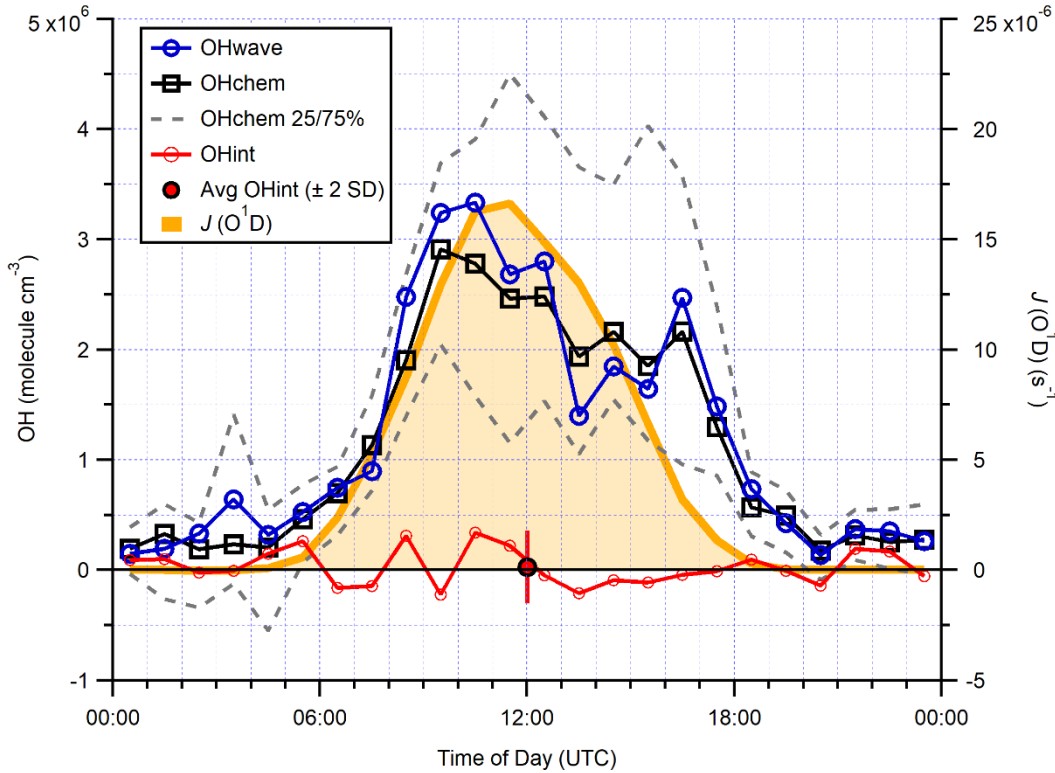

**Figure 11**. Hourly median diurnal profiles of OHwave, OHchem, and $J(O^1D)$ (right axis) from the ICOZA campaign. Also shown (red line and markers) is the hourly median diurnal profile of OHint (= OHwave − OHchem), calculated from individual 4 min data points; the single red marker corresponds to the average ($\pm 2\sigma$) of this trace. The variability (interquartile range, IQR) in OHchem measurements is denoted by the grey dashed lines, not shown for others for clarity. OHwave data were corrected for the known interference from $O_3 + H_2O$.





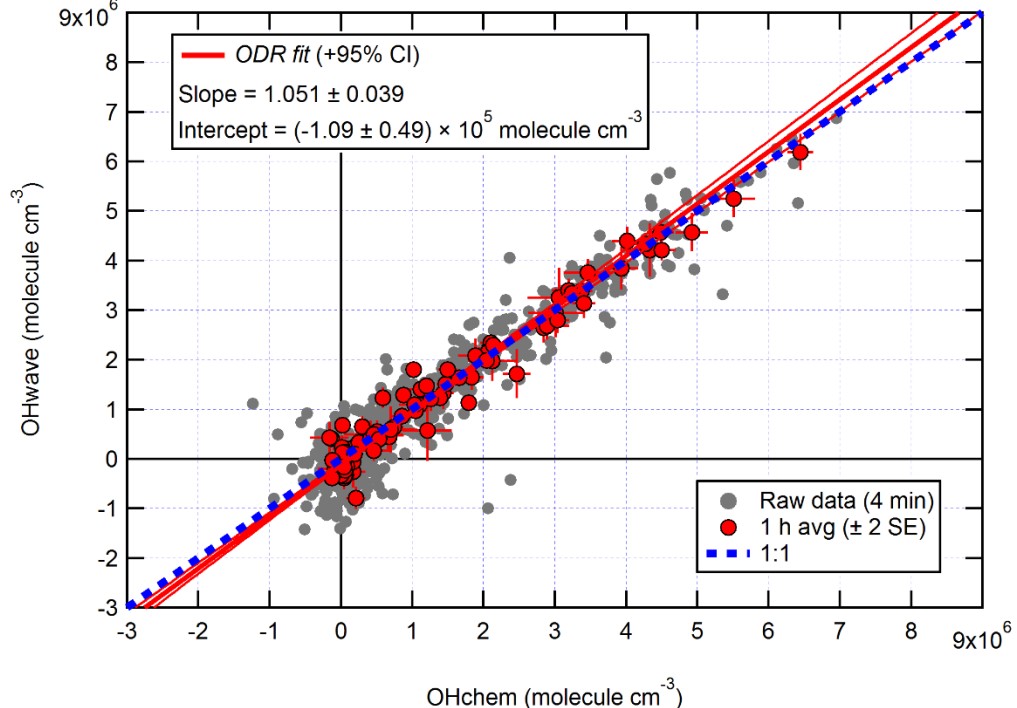

**Figure 12.** Overall intercomparison of OHwave and OHchem observations from the winter 2016 AIRPRO campaign. Grey markers represent raw data (4 min), with 1 h averages (±2 SE) in red. The thick red line is the ODR fit to the hourly data, with its 95% CI bands given by the thin red lines; fit errors given at the $2\sigma$ level. For comparison, 1:1 agreement is denoted by the blue dashed line. OHwave data were corrected for the known interference from $O_3 + H_2O$.

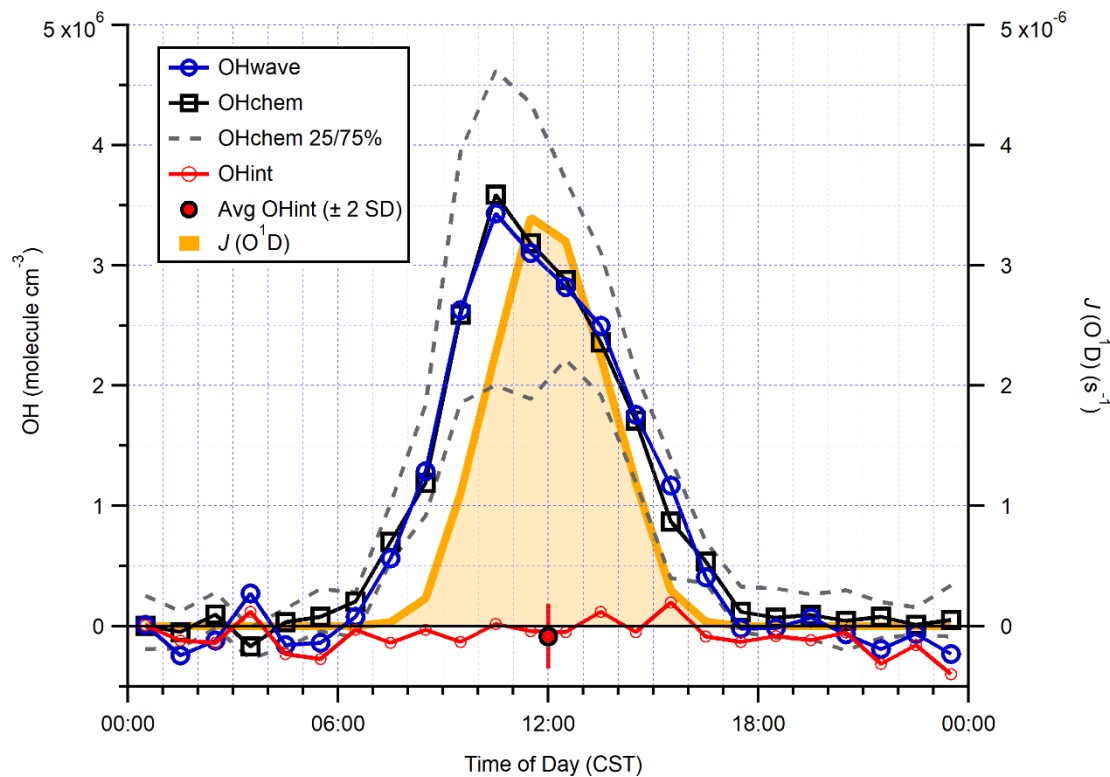

**Figure 13**. Hourly median diurnal profiles of OHwave, OHchem, and $J(O^1D)$ (right axis) from the winter 2016 AIRPRO campaign. Also shown (red line and markers) is the hourly median diurnal profile of OHint, calculated from individual 4 min data points; the single red marker corresponds to the average ($\pm 2\sigma$) of this trace. The variability (IQR) in OHchem measurements is denoted by the grey dashed lines, not shown for others for clarity. OHwave data were corrected for the known interference from $O_3 + H_2O$.





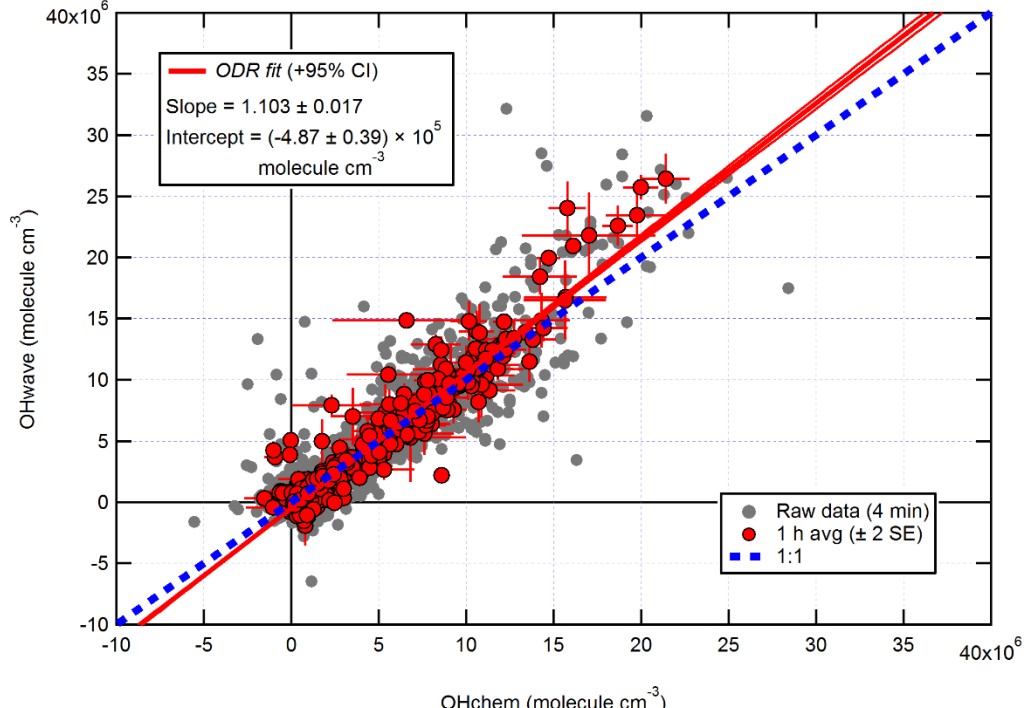

**Figure 14.** Overall intercomparison of OHwave and OHchem observations from the summer 2017 AIRPRO campaign. Grey markers represent raw data (4 min), with 1 h averages (±2 SE) in red. The thick red line is the ODR fit to the hourly data, with its 95% CI bands given by the thin red lines; fit errors given at the $2\sigma$ level. For comparison, 1:1 agreement is denoted by the blue dashed line. OHwave data were corrected for the known interference from $O_3$ + $H_2O$.

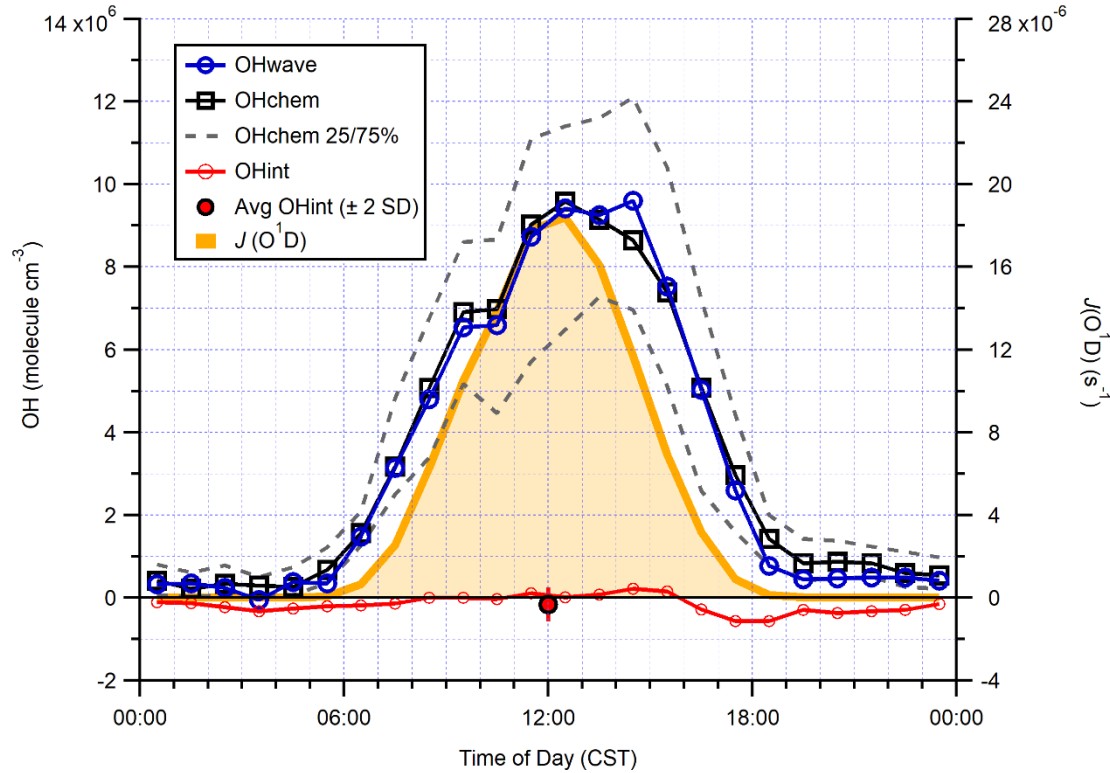

**Figure 15**. Hourly median diurnal profiles of OHwave, OHchem, and $J(O^1D)$ (right axis) from the summer 2017 AIRPRO campaign. Also shown (red line and markers) is the hourly median diurnal profile of OHint, calculated from individual 4 min data points; the single red marker corresponds to the average ($\pm 2\sigma$) of this trace. The variability (IQR) in OHchem measurements is denoted by the grey dashed lines, not shown for others for clarity. OHwave data were corrected for the known interference from $O_3 + H_2O$.

**Figure 16.** Relationship between OHint and OHchem using binned data. The error bars for AIRPRO summer (upper panel, blue markers) denote 1 SD, not shown for AIRPRO winter and ICOZA for clarity. The black dashed line corresponds to a sigmoid fit used to guide the eye only. The contribution of interferences to the total OHwave signal (= OHint/OHwave × 100%) for the AIRPRO summer campaign is shown in the middle panel, with the number of points in each bin shown in the lower panel. All OHint data used here have been corrected for the known interference from $O_3 + H_2O$.