# Peer review of "Implementation of a chemical background method for atmospheric OH measurements by laser-induced fluorescence: characterisation and observations from the UK and China"

_Atmospheric Measurement Techniques, 2019_

## Referee Comment (RC1) · Anonymous Referee #1 · 27 Jan 2020

This manuscript describes OH interference measurements performed using the Leeds LIF FAGE instrument with the inclusion of an inlet pre-injector during three separate field deployments. I was very interested in reviewing this manuscript due to its importance in the measurement of ambient OH. With large reported discrepancies between measured and modeled OH and the significance of accurate OH measurements, it is important for all LIF groups to conduct these tests. This manuscript is thorough and well written; however, I have concerns regarding some of the analyses which I believe can be resolved with further clarification. Nevertheless, once these changes are made,

[Figure]

I fully recommend publication in AMT.

Specific comments

Figure 5: Do the authors have any thoughts on why there is more variability in the propane results? And why does C3F6 produce a more gradual decay?

P11, line 367: I suggest the authors consider adding the description of the injector tip to Figure 1 for clarity. In addition, can it be clarified that this injector tip is not interrupting the sample flow?

P13, lines 464-71: I suggest the authors perform a simple modeling test to determine if SCIs are the likely cause of this interference. While it is negligible when extrapolated to ambient concentrations, acknowledgement and quantification of this interference, if it is in fact from SCIs, can be used in comparison with other FAGE instrumental overviews.

Figure 10: Do the authors have any suggestions as to why there appears to be greater variability at lower OHwave and OHchem measurements? Assuming the majority of these points are nighttime measurements, is NO3 responsible? Overcorrection of O3/H2O?

Section 2.3: More information about the field deployments needs to be presented. How long (#days) were each deployment? Were measurements continuous? The authors later mention instrumental issues such as power outages. This would be the section to provide more detail about such matters.

P 16, lines 544-8: Did the authors compare OHint to the same parameters that were previously implicated? It seems that since there was little to no OH interference measured, that by comparing OHwave and OHchem individually to these parameters, the trend will largely be representative of actual OH.

P16, lines 558-9: Are the authors convinced that an over-subtraction of O3/H2O is not occurring during the daytime as well? Figure 13 shows OHchem larger than OHwave at peak concentrations. In addition, AIRPRO summer 2017 reports a negative interference ratio (-0.09). Is this also related to an over correction of O3/H2O? The authors are encouraged to give more insight into this.

P16, line 573: Do the authors have any suggestions for why the nighttime OH levels were so much higher during AIRPRO summer 2017 in comparison to the other two campaigns?

P17, line 578-87: While there are few points of high OH measurements, was any analysis performed on their relationship with BVOCs, O3, or temperature? Were these high concentrations measured on different days/times? While it may be limited, the authors are suggested to provide more of an analysis of these points.

P18, lines 611-13: The sentence 'These findings. . .in this type of environment.' should be reworded or omitted. While one of the field sites showed high BVOC and low NO concentrations, it should not be implied as being representative of a forested environment. While the BVOC and NO conditions may have been similar, other key compounds, such as SO2, would have likely been larger, altering the environmental conditions further from that of a forest.

Technical corrections

Abstract. P1, line21: change scavenging to scavenger

P3, line 81: either remove the word 'by' or change to 'Mao et al. (2012)'

P3, line 82: Lew et al., 2019 should be added as a reference for OH interference measurements in a forested environment

P5, line 175: add reference Rickly and Stevens, 2018

P8, line 267: move comma to read as 'AIRPRO winter, but after'

Figure 4: make red and blue markers consistent with graph 3

P9, line 303: remove second 'reduction' in this sentence

P12, line 398: change to 'OH removal efficiency of ∼12%'

P16, line 567: change 'as' to 'because'

P17, line 590: either remove 'of' or change to 'Fittschen et al. (2019)'

P17, lines 599-601: The authors are suggested to add 'in moderately to highly polluted environments' to the end of this sentence for clarification.
* * *

---

## Referee Comment (RC2) · Anonymous Referee #2 · 13 Mar 2020

**General**

The paper describes the implementation of a chemical modulation technique for the FAGE-LIF OH instrument operated by the University of Leeds. The purpose of this technical upgrade is to quantify and correct for possible OH interferences, which cannot be detected by the traditionally used laser-excitation wavelength modulation technique. Both operational methods, chemical and wavelength modulation, allow to discriminate OH signals from background that is caused by scattered laser radiation, non-resonant

fluorescence, or solar radiation entering the instrument. However, only chemical modulation, in which ambient OH is scavenged by an added reagent in front of the instrument inlet, allows the detection and subtraction of signals from OH, which is artificially produced inside the instrument. Over the last eight years, some LIF groups have discovered, by use of chemical modulation, previously unknown significant interferences due to instrumental OH, when measurements were performed in biogenically influenced environments. Currently, it is not clear which precursors or formation mechanisms are responsible for the observed artefacts. It is also not clear how much these interferences depend on the specific instrumental design and its operating conditions. The comparison of measured and model simulated OH concentrations in the real atmosphere has always been considered as an important test of our understanding of atmospheric chemistry. It is obvious that progress in understanding can only be achieved if atmospheric OH measurements are reliable. The implementation of the chemical modulation technique in LIF-FAGE instruments as in the present case is therefore an important step in the further development of the research field. The current paper deals with this important topic and is worth to be published in AMT.

The paper provides a good overview of the topic and describes in detail the implementation in the Leeds LIF-FAGE instrument. The authors present interesting results of interference tests in the laboratory and of applications of chemical modulation with the Leeds instrument during field campaigns in the UK and China. The description of the technique and results is clear and well structured. In agreement with laboratory work from other groups, the authors find that OH interferences from ozone photolysis, nitrate radicals and ozonolysis of alkenes are generally negligible at natural atmospheric conditions, or can be sufficiently well corrected (ozone photolysis). In relative clean air in UK and in the polluted air in Beijing, China, the authors find no evidence of unknown OH artefacts larger than the detection limit. This means that either the instrumental design of Leeds is less sensitive to OH artefacts found in other instruments, or that different environmental conditions have not supported the formation of such artefacts. In any case, the application of the new method increases confidence in OH field data

obtained by the Leeds instrument. I recommend publication of the paper after the authors have adressed my comments below.

**Major comments**

Abstract

I am missing a quantitative statement about the possible extent of unknown interferences in the FAGE instrument of Leeds during the AIRPRO and ICOZA campaigns. Please specify upper limits as equivalent OH number densities and fractional contributions to the measured total OH signals (without scavenger) taking into account the measurement errors.

Internal OH removal efficiency

The authors present a clever idea to determine the internal removal efficiency of OH by propane. However, I think the evaluation of the data presented in Table 2 needs some revisions. The average value ($\pm 2\sigma$) for the internal removal is given in the text as $(-0.5 \pm 1.3)$% at a propane mixing ratio of 110 ppmv and $(-2.8 \pm 2.3)$% at 1100 ppmv. I believe that the stated errors are too small. The number of experiments mentioned for both cases (Table 2) is rather low (n=3). The calculated standard deviation from such small statistical sample underestimates the error of the mean value. I suggest to calculate weighted means with their standard errors (using error propagation). This would yield $(0.0 \pm 4.0)$% at 110 ppmv propane and $(2.9 \pm 6.6)$% at 1100 ppmv. While the mean values are not much different from the ones given in Table 2, the larger errors seem more plausible.

An internal OH removal of 12% is theoretically calculated for the case that the sampled OH is exposed to the scavenger (1100 ppmv) for 2 ms on the way from pinhole to laser axis. The experimental value of 2.8% (Table 2) cannot be directly compared with this

theoretical estimate, as is done in the paper. In the instrument, OH is built up gradually (in this case linearly) by $HO_2$ conversion along the line from pinhole to the laserbeam, followed by OH reaction with propane (and NO). In this sequential reaction system, the effective scavenging efficiency is about half the efficiency for OH radicals exposed to the reactant over the entire distance from the inlet to the laser beam. Thus, the experimental value has to be approximately doubled to be comparable with the theoretical estimate. A value of $2\times (2.9 \pm 6.6)\% = (5.8 \pm 13)\%$ would not be much different from the theoretical value of 12%. My conclusion is that 10% internal OH loss at 1100 ppmv of propane cannot be ruled out by these laboratory experiments.

Intercomparison OHwave and OHchem

In Figure 10, 12 and 14, OHwave is higher than OHchem by 16%, 5%, and 10%, respectively. The discrepany is statistically significant (i.e., larger than the $\pm 2\sigma$ statistical errors of the fitted slopes). There must be reasons for the systematic deviations which should be discussed in more detail. I am not satisfied by the statement that the discrepancies can be explained by the instrumental uncertainties (26%, $\pm 2\sigma$). Two measurements are compared which actually use the same calibration. Thus, uncertainties of parameters used to quantify the OH production in the wand cannot be responsible for the differences between OHwave and OHchem. How much of the discrepancy between OHwave and OHchem can be explained by the uncertainty of the $O_3$-$H_2O$ interference correction in OHwave? Are there other possible reasons? Finally, the differences could indicate an uncorrected bias due to an unknown interference in OHwave, which, I agree, would be smaller than the instrumental uncertainty of 26%.

**Minor comments**

Line 125. SI unit should be used for pressure (e.g., hPa) instead of Torr.

Line 149. initially to $HO_2$ and subsequently to OH ?

Line 290-295. It should be explicitly stated that the determination of OHwave and OHchem uses the same calibration, which is carried out without IPI under the assumption of negligible transmission losses with IPI.

Line 290-295. For the given IPI conditions, I calculate a Reynolds number of 2290. This value is close to the critical point where laminar flow becomes turbulent. The state of the flow is expected to influence the mixing of the scavenger in the IPI flow and the loss rate of OH at the walls. Have the authors tested, how sensitive the scavenging efficiency and tube transmission depends on the IPI flow rate?

Line 345. I assume you mean $(0.030 \pm 0.091)$% instead of $0.030 \pm 0.091$% ? Check also other instances in the paper.

Line 387. The number for the internal removal of $-0.5 \pm 1.3$% is not consistent with the value in Table 2 showing $-0.2 \pm 1.1$%.

Line 413. I assume, the water level is given as a volume mixing ratio and not as relative humidy. Please clarify.

Line 435. How long is the reaction time for isoprene and ozone before the gas is entering the pinhole of the FAGE cell?

Line 444. Which material was used for the additional 30cm flow tube?

Line 445. What is the meaning of '$\tau$ = 0.15 s' ?

Table 1, footnotes. Labels are missing in the table body.

Table 3. Column width of 'Obs' needs reformatting.

Table 4. What is the meaning of the $\star$ symbol for the daytime contribution in the CalNex-LA study? Nighttime column: what is the conceptual difference between $\sim 0$ (e.g., PROPHET, AIRPRO summer) and 'Nighttime OH almost always $<$ LOD' (AIRPRO Winter)? The reported OH interferences in the PRIDE-PRD2014 campaign made contributions up to 8% during daytime and up to 20% at sunset and nighttime.

Fig. 4, 6, 7c, and 9. Error bars are much larger than the scatter of the shown data. Therefore, the error bars do not seem to represent the precision of the shown data. If you show mean values of repeat experiments, you may want to display the statistical error of the mean rather than of single measurements.

Figure 1. Insert a scale to illustrate the size of the IPI.

Figure 4. Consider to include the diurnal profile of jO1D scaled to OH; as jO1D and OH often correlate extremely well, it could help to visualise the expected trend of OH while the IPI switches between modes.

Figure 6. For better understanding, you could add in the figure caption the information that OH is internally formed in the cell by the conversion of $HO_2$ with added NO.

Figure 8. The y-axis is labelled 'HOx' cell signal. Please clarify: was the sum of OH and $HO_2$ measured (i.e., with added NO), or only OH (without added NO) ?

---

## Author Comment (AC1) · 6 Apr 2020

**Response to RC1**

This manuscript describes OH interference measurements performed using the Leeds LIF FAGE instrument with the inclusion of an inlet pre-injector during three separate field deployments. I was very interested in reviewing

- 5 this manuscript due to its importance in the measurement of ambient OH. With large reported discrepancies between measured and modeled OH and the significance of accurate OH measurements, it is important for all LIF groups to conduct these tests. This manuscript is thorough and well written; however, I have concerns regarding some of the analyses which I believe can be resolved with further clarification. Nevertheless, once these changes are made, I fully recommend publication in AMT.
- 10 We thank the reviewer for their kind comments. Below we present the comments of the reviewer (blue text), then our responses to individual comments and changes made in the revised manuscript (black text).

**Specific comments**

Figure 5: Do the authors have any thoughts on why there is more variability in the propane results? And why does C3F6 produce a more gradual decay?

15 Some of the propane results were collected when the FAGE instrument was operating with a higher background than usual, leading to more scatter in the data. Considering we are measuring on the order of <1% OH remaining, we would expect some scatter in the data. With this considered it is clear that propane and C3F6 follow the same general trend. Finally, we do not believe this affects our main point that under the IPI operating conditions used during field campaigns, >99% of ambient OH should have been scavenged away.

20

P11, line 367: I suggest the authors consider adding the description of the injector tip to Figure 1 for clarity. In addition, can it be clarified that this injector tip is not interrupting the sample flow?

We have added "through 1/8" stainless steel tubing" after "injected" in the revised MS. We cannot clarify that the sample flow is interrupted by the injectors and see this as a limitation to our approach for determining internal OH

removal. Since the NO injector tip was only moved for the purpose of the internal removal experiment, we have chosen not to add this to Fig. 1 (in ambient HO2 detection, NO is injected 7.5 cm below the pinhole). P13, lines 464-71: I suggest the authors perform a simple modeling test to determine if SCIs are the likely cause of this interference. While it is negligible when extrapolated to ambient concentrations, acknowledgement and quantification of this interference, if it is in fact from SCIs, can be used in comparison with other FAGE instrumental overviews.

- 5 We agree with the reviewer that modelling the SCI decomposition and quantification of the interference would be useful for comparison to other FAGE instruments. We have added the following sentences to Section 3.2.2: "Similarly, we have modelled the SCI decomposition in our FAGE cell. Assuming an ambient atmosphere containing 100 ppbv  $O_3$  and 10 ppbv alkene and taking the reactions and rate coefficients from Novelli et al. (2014a), we calculate an equivalent ambient pressure OH concentration of ~4 × 103 molecule cm-3 from the
- 10 decomposition of SCIs at our FAGE cell residence time of 2 ms." Novelli, A., Vereecken, L., Lelieveld, J., and Harder, H.: Direct observation of OH formation from stabilised Criegee intermediates, Phys. Chem. Chem. Phys., 16, 19941–19951, 2014.

Figure 10: Do the authors have any suggestions as to why there appears to be greater variability at lower OHwave

15 and OHchem measurements? Assuming the majority of these points are nighttime measurements, is NO3 responsible? Overcorrection of O3/H2O?

We suggest the perceived greater variability at low OH is simply due to more datapoints being collected at low OH, thus making it more likely to see outliers. NO3 interferences have been ruled out as insignificant in laboratory tests, and we believe the  $O_3/H_2O$  interference has been well quantified, although since this is very small it is difficult

20 to measure accurately.

Section 2.3: More information about the field deployments needs to be presented. How long (#days) were each deployment? Were measurements continuous?

The text has been modified accordingly.

25 ICOZA: "Two continuous IPI sampling periods were conducted in the middle of the campaign, separated by a few days (3rd-8th July and 12th-16th July), with a total of nine days where OHchem measurements are available around midday. For other times, only measurements of OHwave are available. During the IPI sampling periods, power cuts on the nights of 3rd/4th July and 6th/7th July resulted in data loss."

AIRPRO: "In winter, OHwave and OHchem were measured simultaneously for 6 days of the campaign. In summer,

30 almost one month of near-continuous IPI data are available, with one day of interruptions due to IPI testing (see Section 3.1.2)." The authors later mention instrumental issues such as power outages. This would be the section to provide more detail about such matters.

We have added the following sentence to the text referring to ICOZA: "During the IPI sampling periods, power cuts on the nights of 3rd/4th July and 6th/7th July resulted in data loss."

P 16, lines 544-8: Did the authors compare OHint to the same parameters that were previously implicated? It seems that since there was little to no OH interference measured, that by comparing OHwave and OHchem individually to these parameters, the trend will largely be representative of actual OH.

10 We agree with this comment and have looked at correlations of OHint with other species. For ICOZA, no dependences were found on parameters previously implicated:

**Figure 1.** Daytime  $(J(O^1D) > 5e-7)$  OHint binned against various parameters for the ICOZA campaign. Error bars correspond to 1 SD.

The case was similar for AIRPRO summer:

5

**Figure 2.** Daytime  $(J(O^1D) > 5e-7)$  OHint binned against various parameters for the summer AIRPRO campaign. Error bars correspond to 1 SD.

Although, it can be seen that OHint was marginally higher in the highest temperature, JO1D, and isoprene bins. We have decided to included these figures in a supplementary information file, and referenced them in the main MS.

5 P16, lines 558-9: Are the authors convinced that an over-subtraction of O3/H2O is not occurring during the daytime as well? Figure 13 shows OHchem larger than OHwave at peak concentrations. In addition, AIRPRO summer 2017 reports a negative interference ratio (-0.09). Is this also related to an over correction of O3/H2O? The authors are encouraged to give more insight into this.

We believe the O3/H2O interference has been well quantified, although since this is very small it is difficult to 10 measure accurately (see, e.g., the high scatter in Figure 7). The scaling factor in equation (E7) has a  $2\sigma$  uncertainty of 140 molecule cm-3 ppbv-1 %-1 mW-1. Subtracting this from the actual factor (520 molecule cm-3 ppbv-1 %-1 mW-1) yields a lower limit for the O3/H2O interference. Below shows the effect of using this lower limit interference to correct OHwave data, resulting in less negative OHint. However, in both the original and lower limit cases, OHint is still more negative than the 1 h LOD in the afternoon. OHint levels less negative than the 1 h LOD can only be achieved by reducing the O3/H2O interference to zero. It is not clear why this is the case. Since the O3/H2O had to be determined at very high O3, one possibility is that, in our instrument, the dependence is nonlinear at ambient O3 levels.

Figure 3. Effect of reducing the known interference from O3/H2O.

P16, line 573: Do the authors have any suggestions for why the nighttime OH levels were so much higher during AIRPRO summer 2017 in comparison to the other two campaigns?

- Interpretation of the chemistry is beyond the scope of this work but will be discussed in detail in forthcoming manuscripts. It is likely that the AIRPRO summer campaign had much more active NO3 and ozonolysis chemistry due to high nighttime NO2, O3, and NO3, leading to increased nighttime radical production. In other field campaigns in China, significant nighttime OH concentrations have also been found. For example, in Lu et al. (2014), nighttime OH was up to  $3 \times 10^6$  molecule cm-3, which is higher than the nighttime OH in our work. To reconcile such high
- OH levels required the inclusion of additional ROx production processes in their model.
  Lu, K. D., Rohrer, F., Holland, F., Fuchs, H., Brauers, T., Oebel, A., Dlugi, R., Hu, M., Li, X., Lou, S. R., Shao, M., Zhu, T., Wahner, A., Zhang, Y. H., and Hofzumahaus, A.: Nighttime observation and chemistry of HOx in the Pearl River Delta and Beijing in summer 2006, Atmos. Chem. Phys., 14, 4979–4999, https://doi.org/10.5194/acp-14-4979-2014, 2014.

20

5

P17, line 578-87: While there are few points of high OH measurements, was any analysis performed on their relationship with BVOCs, O3, or temperature? Were these high concentrations measured on different days/times? While it may be limited, the authors are suggested to provide more of an analysis of these points.

Please see our relevant response above. For AIRPRO summer, no clear dependence of OHint was found on isoprene,  $O_3$ , or temperature.

P18, lines 611-13: The sentence 'These findings. . .in this type of environment.' should be reworded or omitted. While one of the field sites showed high BVOC and low NO concentrations, it should not be implied as being representative of a forested environment. While the BVOC and NO conditions may have been similar, other key

10 compounds, such as SO2, would have likely been larger, altering the environmental conditions further from that of a forest.

We have reworded the sentence to: "Although AIRPRO summer took place in a city, its results do provide confidence in previous measurements of OH using the same instrument, and support the hypothesis that there are unknown OH sources in the atmosphere."

**15 Technical corrections**

Abstract. P1, line 21: change scavenging to scavenger Done

P3, line 81: either remove the word 'by' or change to 'Mao et al. (2012)'

**20 Done**

5

P3, line 82: Lew et al., 2019 should be added as a reference for OH interference measurements in a forested environment

**Done**

**25**

P5, line 175: add reference Rickly and Stevens, 2018

**Done**

P8, line 267: move comma to read as 'AIRPRO winter, but after'

**Done**

Figure 4: make red and blue markers consistent with graph 3 Done

**5**

P9, line 303: remove second 'reduction' in this sentence Done

P12, line 398: change to 'OH removal efficiency of ~12%'

**10 Done**

P16, line 567: change 'as' to 'because' Done

P17, line 590: either remove 'of' or change to 'Fittschen et al. (2019)'Done

P17, lines 599-601: The authors are suggested to add 'in moderately to highly polluted environments' to the end of this sentence for clarification.

20 The sentence has been changed to: "The results from the three field campaigns that feature in this work demonstrate that, in moderately to highly polluted conditions, the Leeds ground-based FAGE instrument does not suffer from substantial interferences in the measurement of OH using the conventional, wavelength-modulation background technique, OHwave."

---

## Author Comment (AC2) · 6 Apr 2020

**Response to RC2**

**General**

The paper describes the implementation of a chemical modulation technique for the FAGE-LIF OH instrument operated by the University of Leeds. The purpose of this technical upgrade is to quantify and correct for possible OH interferences, which cannot be detected by the traditionally used laser-excitation wavelength modulation technique. Both operational methods, chemical and wavelength modulation, allow to discriminate OH signals from background that is caused by scattered laser radiation, non-resonant fluorescence, or solar radiation entering the instrument. However, only chemical modulation, in which ambient OH is scavenged by an added reagent in front of the instrument inlet, allows the detection and subtraction of signals from OH, which is artificially produced inside the instrument. Over the last eight years, some LIF groups have discovered, by use of chemical modulation, previously unknown significant interferences due to instrumental OH, when measurements were performed in biogenically influenced environments. Currently, it is not clear which precursors or formation mechanisms are responsible for the observed artefacts. It is also not clear how much these interferences depend on the specific instrumental design and its operating conditions. The comparison of measured and model simulated OH concentrations in the real atmosphere has always been considered as an important test of our understanding of atmospheric chemistry. It is obvious that progress in understanding can only be achieved if atmospheric OH measurements are reliable. The implementation of the chemical modulation technique in LIF-FAGE instruments as in the present case is therefore an important step in the further development of the research field. The current paper deals with this important topic and is worth to be published in AMT.

The paper provides a good overview of the topic and describes in detail the implementation in the Leeds LIF-FAGE instrument. The authors present interesting results of interference tests in the laboratory and of applications of chemical modulation with the Leeds instrument during field campaigns in the UK and China. The description of the technique and results is clear and well structured. In agreement with laboratory work from other groups, the authors find that OH interferences from ozone photolysis, nitrate radicals and ozonolysis of alkenes are generally negligible at natural atmospheric conditions, or can be sufficiently well corrected (ozone photolysis). In relative clean air in UK and in the polluted air in Beijing, China, the authors find no evidence of unknown OH artefacts larger than the detection limit. This means that either the instrumental design of Leeds is less sensitive to OH

artefacts found in other instruments, or that different environmental conditions have not supported the formation of such artefacts. In any case, the application of the new method increases confidence in OH field data obtained by the Leeds instrument. I recommend publication of the paper after the authors have adressed my comments below.

We thank the reviewer for their kind comments. Below we present the comments of the reviewer (blue text), then

5  our responses and any changes in the revised manuscript (black text).

**Major comments**

**Abstract**

I am missing a quantitative statement about the possible extent of unknown interferences in the FAGE instrument of Leeds during the AIRPRO and ICOZA campaigns. Please specify upper limits as equivalent OH number

10  densities and fractional contributions to the measured total OH signals (without scavenger) taking into account the measurement errors.

There are a number of ways to estimate an upper limit interference from the data we have presented. One approach would be to take the maximum value of OHint. For example, the maximum 1 h OHint was $2.4 \times 10^6$ molecule cm$^-$$^3$, accounting for 67% of the total OHwave signal at this time. However, 1 h OHint values were scattered around

15  zero such that the minimum OHint ($-1.7 \times 10^6$ molecule cm$^{-3}$) had almost the same absolute magnitude. For this reason, we do not think the above method is a fair way to determine upper limit interferences.

Therefore, we have chosen to estimate the upper limit interference from the data presented in Fig. 16. We have added the following sentence as the last sentence in the abstract: "The difference between OHwave and OHchem ("OHint") was found to scale nonlinearly with OHchem, resulting in an upper limit interference of $(5.0 \pm 1.4) \times$

20  $10^6$ molecule cm$^{-3}$ at the very highest OHchem concentrations measured ($23 \times 10^6$ molecule cm$^{-3}$), accounting for ~15–20% of the total OHwave signal."

Upper limit fractional contributions (but not absolute number densities) can also be derived from the fits in Figs 10, 12, and 14. The positive 2σ limits of the slopes (i.e., slope + uncertainty) were 1.22, 1.09, and 1.12 for ICOZA, AIRPRO winter, and AIRPRO summer respectively, i.e., upper limit contributions of ~10–20%.

25

**Internal OH removal efficiency**

The authors present a clever idea to determine the internal removal efficiency of OH by propane. However, I think the evaluation of the data presented in Table 2 needs some revisions. The average value (±2σ) for the internal removal is given in the text as (−0.5 ± 1.3)% at a propane mixing ratio of 110 ppmv and (−2.8 ± 2.3)% at 1100 ppmv. I believe that the stated errors are too small. The number of experiments mentioned for both cases (Table 2) is rather low (n=3). The calculated standard deviation from such small statistical sample underestimates the error of the mean value. I suggest to calculate weighted means with their standard errors (using error propagation). This would yield (0.0 ± 4.0)% at 110 ppmv propane and (2.9 ± 6.6)% at 1100 ppmv. While the mean values are not much different from the ones given in Table 2, the larger errors seem more plausible.

We agree with the method of error estimation suggested by the reviewer and have amended Table 2 accordingly.

An internal OH removal of 12% is theoretically calculated for the case that the sampled OH is exposed to the scavenger (1100 ppmv) for 2 ms on the way from pinhole to laser axis. The experimental value of 2.8% (Table 2) cannot be directly compared with this theoretical estimate, as is done in the paper. In the instrument, OH is built up gradually (in this case linearly) by HO2 conversion along the line from pinhole to the laserbeam, followed by OH reaction with propane (and NO). In this sequential reaction system, the effective scavenging efficiency is about half the efficiency for OH radicals exposed to the reactant over the entire distance from the inlet to the laser beam. Thus, the experimental value has to be approximately doubled to be comparable with the theoretical estimate. A value of 2× (2.9 ± 6.6)% = (5.8 ± 13)% would not be much different from the theoretical value of 12%. My conclusion is that 10% internal OH loss at 1100 ppmv of propane cannot be ruled out by these laboratory experiments.

We thank the reviewer for pointing out the flaws in our internal removal experiment and how to better compare the measured and theoretical internal removal. We have added the following discussion to Section 3.1.3:

"In the internal removal experiment, OH is not formed instantly at the pinhole but is built up linearly by $HO_2$/NO conversion along the line from the pinhole to the laser axis. Therefore, the experimental internal removal may not be directly compared with the theoretical estimate. In such a sequential reaction system, the OH scavenging is about half as efficient as that for the case where OH is formed as an instant point source at the pinhole. Thus, the experimental value should be doubled to (5.8 ± 13)%, which is in reasonable agreement with the theoretical value. From this, we cannot rule out a small internal OH removal on the order of 10% at the higher propane level used for

the AIRPRO summer campaign. However, no such corrections were applied to the ambient data featured in this work."

Intercomparison OHwave and OHchem

5 In Figure 10, 12 and 14, OHwave is higher than OHchem by 16%, 5%, and 10%, respectively. The discrepany is statistically significant (i.e., larger than the ±2σ statistical errors of the fitted slopes). There must be reasons for the systematic deviations which should be discussed in more detail. I am not satisfied by the statement that the discrepancies can be explained by the instrumental uncertainties (26%, ±2σ). Two measurements are compared which actually use the same calibration. Thus, uncertainties of parameters used to quantify the OH production in

10 the wand cannot be responsible for the differences between OHwave and OHchem. How much of the discrepancy between OHwave and OHchem can be explained by the uncertainty of the O3-H2O interference correction in OHwave? Are there other possible reasons? Finally, the differences could indicate an uncorrected bias due to an unknown interference in OHwave, which, I agree, would be smaller than the instrumental uncertainty of 26%.

We agree with the reviewer in that, although the slopes in Figs 10, 12, and 14 indicate small contributions from
15 interferences, we cannot rule unknown interferences out since the slopes are all significantly greater than 1. We believe the O3/H2O interference has been well characterised and thus is not the reason for the differences seen between OHwave and OHchem. Therefore, the differences must be due to unknown interferences, although these are smaller than the instrumental uncertainty of 26% at 2σ.

To accommodate the reviewer's suggestions, we have amended the first paragraph in the discussion section to:

20 "The results from the three field campaigns that feature in this work demonstrate that, in moderately to highly polluted conditions, the Leeds ground-based FAGE instrument does not suffer from substantial interferences in the measurement of OH using the conventional, wavelength-modulation background technique, OHwave. This is illustrated best by the slopes of the overall measurement intercomparison plots (Figures 10, 12, and 14), which ranged from 1.05–1.16. However, while the deviations of these slopes from 1 are small, they are still significant,
25 suggesting the presence of unknown OH interferences. Nonetheless, such unknown interferences are well within the instrumental uncertainty of ~26% at $2\sigma$."

**Minor comments**

Line 125. SI unit should be used for pressure (e.g., hPa) instead of Torr.

Done

5  Line 149. initially to HO2 and subsequently to OH ?

In the ROxLIF flow tube, RO2 is converted all the way to OH using NO, then reconverted back to HO2 using CO. Inside the FAGE cell, the HO2 is then converted once more to OH using NO. To clarify this, we have modified the sentence as follows:

"Although not reported here, $RO_2$ radicals are measured using the $RO_xLIF$ method (Fuchs et al., 2008; Whalley et 10  al., 2013), in which their reactions with NO and CO (BOC, 5% in $N_2$ and Messer, 10% in $N_2$) result in conversion initially to OH (using NO; $RO_2 \rightarrow HO_2 \rightarrow OH$) and subsequently back to $HO_2$ (using CO; $OH \rightarrow HO_2$) that is then detected as described above (via addition of NO inside the FAGE cell; $HO_2 \rightarrow OH$)."

Line 290-295. It should be explicitly stated that the determination of OHwave and OHchem uses the same 15  calibration, which is carried out without IPI under the assumption of negligible transmission losses with IPI.

We have added the following sentence to Section 3.1.1:

"In other words, we assume negligible transmission losses within the IPI and the OH calibration factor we applied to ambient data was the same for (1) OHwave without IPI sampling, (2) OHwave during IPI sampling, and (3) OHchem during IPI sampling. However, it should be noted that in the field, calibrations are normally carried out 20  without the IPI system present."

Line 290-295. For the given IPI conditions, I calculate a Reynolds number of 2290. This value is close to the critical point where laminar flow becomes turbulent. The state of the flow is expected to influence the mixing of the 25  scavenger in the IPI flow and the loss rate of OH at the walls. Have the authors tested, how sensitive the scavenging efficiency and tube transmission depends on the IPI flow rate?

In preliminary experiments, it was found that the OH transmission through the IPI increased with sheath flow rate (range: 0–25 slm) until the air sent through the wand (max 40 slm) was no longer sufficient to overflow the inlet (sheath flow = 25 slm, total flow through IPI = 32 slm). The increase in transmission with flow is likely due to reduced contact with the walls at faster flow, despite the increased turbulence. Since we observed maximum transmission at the limits of our experimental setup, we settled on those experimental conditions for all future experiments and ambient studies.

Similarly, although we did not test whether the sheath flow rate affected scavenging efficiency, we did test the N2 dilution flow. Again, the maximum scavenging efficiency was observed at the maximum of the MFC used (500 sccm), so we used these flow conditions for all further tests.

Line 345. I assume you mean (0.030±0.091)% instead of 0.030±0.091% ? Check also other instances in the paper.

All instances corrected

Line 387. The number for the internal removal of $-0.5 \pm 1.3\%$ is not consistent with the value in Table 2 showing $-0.2 \pm 1.1\%$.

This was corrected to $(0.0 \pm 4.4)\%$ based on the discussion of weighted averages above.

Line 413. I assume, the water level is given as a volume mixing ratio and not as relative humidy. Please clarify.

We mean volume mixing ratio, this has been clarified.

Line 435. How long is the reaction time for isoprene and ozone before the gas is entering the pinhole of the FAGE cell?

Line 444. Which material was used for the additional 30cm flow tube?

Line 445. What is the meaning of '$\tau = 0.15$ s' ?

We have amended the relevant paragraph to (changes highlighted):

"To test for interferences from isoprene (ISO) ozonolysis products, isoprene (~16 ppmv) and ozone (~1.8 ppmv) were mixed in the calibration wand and the scavenger (propane, PROP) was injected into the IPI flow tube. The propane concentrations were set to those used for ambient OHchem measurements, such that the tests were representative of normal atmospheric sampling (i.e., to test whether an interference signal would remain in ambient data). However, to generate sufficient OH signal for quantitative analysis, ozone and isoprene were introduced at concentrations that far exceeded their typical ambient levels (Table 3). Unlike previous tests of interferences from alkene ozonolysis (Novelli et al., 2014b), low [$O_3$]:[ISO] ratios were used to suppress the signal contribution from the atmospheric (real) OH generated by ozonolysis (i.e., isoprene acted as an additional OH scavenger). To allow sufficient time for steady-state conditions to develop, the IPI did not sample from the calibration wand directly, but instead a 30 cm flow tube (polycarbonate, ID ~ 19 mm) was used to extend the IPI (which sampled wand gas at the normal IPI flow rate of ~32 slm, residence time for $O_3$ + isoprene reaction ~ 0.15 s)."

Table 1, footnotes. Labels are missing in the table body.

Amended

Table 3. Column width of 'Obs' needs reformatting.

Table reformatted

Table 4. What is the meaning of the * symbol for the daytime contribution in the CalNexLA study?

The * symbol means that the interference was consistent with the known O3/H2O interference; we have added a footnote to clarify this.

Nighttime column: what is the conceptual difference between ~ 0 (e.g., PROPHET, AIRPRO summer) and 'Nighttime OH almost always < LOD' (AIRPRO Winter)?

For this column in the table, we put values of ~ 0 for campaigns in which OH was measured above the LOD at night, but OHwave and OHchem were virtually the same. For AIRPRO winter, this was not the case and so it cannot be explicitly stated that the nighttime contribution was 0.

The reported OH interferences in the PRIDE-PRD2014 campaign made contributions up to 8% during daytime and up to 20% at sunset and nighttime.

The daytime contribution has been amended to <8%.

Fig. 4, 6, 7c, and 9. Error bars are much larger than the scatter of the shown data. Therefore, the error bars do not seem to represent the precision of the shown data. If you show mean values of repeat experiments, you may want to display the statistical error of the mean rather than of single measurements.

The error bars in Figs. 4, 6, 7 (a, b, and c) and 9 have all been reduced from 2σ to 1σ to better reflect the apparent precision of the data. Please note, that since weighted fits were used in Fig. 7, this has changed the slope in Fig. 7A and hence the scaling factor in equation (E7).

Figure 1. Insert a scale to illustrate the size of the IPI.

Scale added

Figure 4. Consider to include the diurnal profile of jO1D scaled to OH; as jO1D and OH often correlate extremely well, it could help to visualise the expected trend of OH while the IPI switches between modes.

jO1D profile added

Figure 6. For better understanding, you could add in the figure caption the information that OH is internally formed in the cell by the conversion of HO2 with added NO.

Caption amended to:

Time series of the LIF signal during internal OH removal experiments. The raw 1 s data are given by the grey line. NO was continuously added to the FAGE cell during these experiments (to form OH internally), and points where propane was added to the IPI flow tube are indicated by the orange shaded panels, with the corresponding signal averages ($\pm 1\sigma$) shown as markers (see text for details). The first experiment (left-hand side) corresponds to the propane mixing ratio used for ICOZA, while the second (right-hand side) corresponds to that used for AIRPRO summer. The results of the internal OH removal experiments are summarised in Table 2.

Figure 8. The y-axis is labelled 'HOx' cell signal. Please clarify: was the sum of OH and HO2 measured (i.e., with added NO), or only OH (without added NO) ?

NO was not added in these experiments. The HOx label is used to discriminate the fluorescence cell normally used for ambient OH and HO2 measurements from the reference cell. Since HO2 is not relevant to this experiment, we have modified the label to "OH cell".

---

## Author Comment (AC3) · 6 Apr 2020

Figures for the Response to Referee #1 RC1

[Figure]

[Figure]

**Figure 1.** Daytime ($J(O^1D) > 5e-7$) OHint binned against various parameters for the ICOZA campaign. Error bars correspond to 1 SD.

**Fig. 1.**

[Figure]

[Figure]

**Figure 2.** Daytime ($J(O^1D) > 5e-7$) OHint binned against various parameters for the summer AIRPRO campaign.
Error bars correspond to 1 SD.

**Fig. 2.**

[Figure]

Figure 3. Effect of reducing the known interference from O3/H2O

**Fig. 3.**